# Assessing Typhoon-induced compound flood drivers: a case study in Ho Chi Minh City, Vietnam

Francisco Rodrigues do Amaral[1], Nicolas Gratiot[1,2], Thierry Pellarin[1], and Tran Anh Tu[2,3]

[1]IGE, Univ. Grenoble Alpes, CNRS, IRD, 38000, Grenoble, France
[2]Centre Asiatique de Recherche sur l'Eau (CARE), Ho Chi Minh City University of Technology (HCMUT), VNU-HCM, 268 Ly Thuong Kiet Street, District 10, Ho Chi Minh City, Viet Nam
[3]Vietnam National University-Ho Chi Minh City (VNU-HCM), Thu Duc City, Ho Chi Minh City, Viet Nam

**Correspondence:** Francisco Rodrigues do Amaral (francisco.amaral@univ-grenoble-alpes.fr)

**Abstract.** We investigate the most severe rainfall event ever experienced in Ho Chi Minh City (HCMC), Vietnam. It occurred on November 25th, 2018 when Typhoon (TY) Usagi directly hit HCMC. During this event, there was more than 300 mm in rainfall over 24h which led to flooding and considerable material damages. We propose an in-depth study of TY-induced, compound flood drivers at a short time scale by focusing on the days before and after the event. We use a set of data analysis and signal processing tools to characterize and quantify both coastal and inland effects on the hydrosystem. We found that TY Usagi made landfall without forming a significant storm surge. The extreme rainfall does not translate in immediate river discharge but presents a 16 hour time lag between peak precipitation and peak residual discharge. Nevertheless, increased river water levels can be seen at both urban and upstream stations with a similar time lag. At the upstream river station, residual discharge represents 1.5 % of available rain water and evidence of upstream wide spread flooding was found. At the urban river station, we assess the potential surface run off during the event to be 8.9 % of the upstream residual discharge. However, a time lag in peak river water level and peak rainfall was found and attributed to the combination of high tide and impervious streets which prevented the evacuation of rain water and resulted in street flooding of up to 0.8 m. Overall, it was found that despite not having a significant storm surge, the coastal tidal forcing is the predominant compound flood driver even during severe, heavy rainfall with tidal fluctuations of river water level and respective discharge much larger than the residuals.

## 1 Introduction

Potential impacts of flooding events on low elevation coastal zones (LECZ) are increasing as populations grow and mean sea levels rise. Instances of flooding within coastal delta areas can arise from various underlying physical factors. These factors encompass storm surges, waves, tides, precipitation, and high river discharge (Paprotny et al., 2020). Additionally, the convergence of these elements can lead to flood events known as compound floods, potentially intensifying the overall flood risk (Zscheischler et al., 2018; Yang et al., 2021). Along tropical and sub-tropical regions, the most energetic forcing agents for the coast are storm systems that associate extreme winds with extreme rainfall and that result in severe physical and human damages (Haigh et al., 2014; Gori et al., 2020). Under the impact of global warming and climate change, concerns about such

extreme weather events and their local impacts have increasingly been under the scope of scientists and governments (Haigh et al., 2014; Heidarzadeh et al., 2018; Le et al., 2019).

Vietnam lies within the most active cyclogenesis region in the world (Gao et al., 2020; Feng et al., 2021; Klotzbach et al., 2022) and 4 to 6 TYs (TYs) hit the coast every year from October to December (Thuan et al., 2016). TYs are major drivers of compound flooding as they often produce strong onshore winds and low barometric pressure which cause extreme storm surges and simultaneously generate heavy rainfall (Gori et al., 2020; Couasnon et al., 2020). The southern part of Vietnam has generally been perceived to be less vulnerable to TYs compared to other parts of the country. However, a substantial amount

of TYs has approached southern Vietnam in the past even though the probability of occurrence is smaller than in the northern regions (Takagi et al., 2014). Additionally, a recent study by Wood et al. (2023) simulated 10,000 years of synthetic tropical cyclone activity in the East Sea of Vietnam (also known as South China Sea) and found that storm surges in both northern and southern Vietnamese coastlines would potentially increase by up to 1 m in the coming decades.

The south of Vietnam consists of a LECZ associated with Ho Chi Minh City (HCMC): a mega city with a population density

that can reach up to 30,000 inhabitants/$km^2$ in the urban city center (Nguyen et al., 2019b) (for comparison, Paris has a density of about 20,000 inhabitants/$km^2$). Water is ubiquitous in HCMC which is traversed by the tidal river Saigon - a complex system with a network of about 800 km of watercourses and canals (Ngoc et al., 2016). Additionally, 90 % of the urban area of HCMC is impermeable and leads to strong effects on the hydrological cycle (Vachaud et al., 2020). Furthermore, this leads to recurrent flooding either of riverine origin or by rainfall runoff which coupled with an increase in occurrence of extreme

rainfall events over HCMC (Ho et al., 2014) makes it one of the most vulnerable coastal regions in the world. In particular, flooding vulnerability is linked to sea level rise, rainfall intensification and ground subsidence, which can reach 0.02 m/year (in some geological areas), while 65% of the city is located at less than 1.5 m above sea level (Lossouarn et al., 2016). In addition, HCMC has a population growth rate of about 3% per year with these risks posing a threat to many livelihoods. The urban growth rate (about 16 km$^2$ per year since 2000) is also an important factor as imperviousness of soils reduces infiltration

potential and further increases the flood risk (Hanson et al., 2011; Lossouarn et al., 2016).

On November 25th 2018, HCMC was hit by TY Usagi. According to the forecast department of the Southern Centre for Hydro-Meteorological Forecasting (SCHMF), this was the highest ever recorded rainfall in a 24-hour period and caused widespread urban flooding. In Camenen et al. (2021) a monthly evaluation of the Saigon River's response to this extreme rainfall yielded a paradoxical result: a lack of direct response in both river water level and discharge. However, there is a clear

interest in better understanding how the hydrosystem physically behaved during and just after this event at finer scales than the monthly average. Additionally, assessing the drivers of the resulting compound flood event over HCMC can provide vital insights for research on coastal and urban flooding prevention.

The presence of concurrent or closely sequential instances of extreme rainfall, extreme river discharge and storm surge can result in extensive destruction, surpassing the impact that these events would have individually (Camus et al., 2021; Eilander

et al., 2023; Heinrich et al., 2023). Numerous studies conducted in recent years have highlighted the significance and highly destructive characteristics of compound flood occurrences connected to TY events across diverse geographical areas (Ye et al., 2021; Yang et al., 2021). A frequent approach is to use statistical models such as copula-based analysis (Ai et al., 2018; Xu

et al., 2022a) or Monte-Carlo approaches (Heinrich et al., 2023) to assess the dependence between TY induced flood drivers in order to understand the likelihood of co-occurring drivers. Statistical methods generally offer the advantage of requiring relatively moderate computational resources. However, this advantage is counterbalanced by the use of meteorological inputs that frequently lack the necessary spatial resolution in tropical areas to accurately encompass the influences of cyclone activity on storm surge (Cid et al., 2018). Another approach is the application of hydrodynamic model simulations to comprehend the intricate physical interplay among driving factors and their respective significance in influencing the overall flood risk (Gori et al., 2020; Liu et al., 2022). Modeling storm surges induced by TYs on a continental or global scale presents a formidable challenge. This is primarily due to the fact that these highly intense storms often possess diameters that fall below the resolution of the model mesh or are attenuated within the broad grid cells of meteorological datasets (Wood et al., 2023). Another downside of modelling approaches is that validation efforts require good quality field measurements which are hard to obtain in many regions of the world. Furthermore, estuarine regions are areas of active morphological changes and thus, in-situ measurements (such as bathymetry) are dynamic and might not accurately represent reality within few years time.

In order to investigate compound flood events within a statistical framework it is ideally necessary to have long time series data (50–100 years) that are both spatially and temporally coherent, encompassing daily river discharge and coastal sea level measurements. Moreover, it's important to note that the availability of daily data varies significantly among rivers, even within regions characterized by relatively good data coverage and is especially scarce in tropical regions (Wood et al., 2023; Scheiber et al., 2023). These limitations can be partially offset by using model-generated data (Xu et al., 2022a; Heinrich et al., 2023) or satellite data. One example is provided by Zhang and Najafi (2020) who model compound flooding caused by tropical storm Matthew over the small Caribbean island of Saint Lucia. As it is data-scarce region there were no in-situ records during the event. Thus, they relied on satellite-based inundation images to validate hydrological and hydrodynamic models. However, a sensitivity analysis showed that uncertainties in flooding are significant in areas close to the coast and thus a probabilistic approach to understand the chances of flooding is required.

Past studies of compound flood events focus mainly on statistical inter-dependency of rainfall and coastal water levels to provide chances of occurrence, conceptual risk models or inundation maps. Furthermore, conceptual risk models are defined in a per-case basis and might not be globally applicable. Ai et al. (2018) present a disaster risk model where rain storms and TY surges are assumed to equally affect flood risk. Additionally, the thresholds used to consider precipitation and or storm surge events as extreme is up to the individual authors and it depends on the region under scope. Moreover, case studies that focus on synthetic events such as the ones performed by Xu et al. (2022a) and Torres-Freyermuth et al. (2022) do not present a connection between these statistical and model outputs to in-situ observations of flood depths and locations usually due to their unavailability. Further, compound flood assessments due to storm surge are usually studied with relation to either rainfall or river discharge, and thus, the combined effects of these two drivers is not investigated.

In this paper, we propose an in-depth assessment of the drivers of compound flooding in this urban estuary region by mainly focusing on in-situ observations during the period around TY Usagi. This is done in a data-scarce region where such data is usually hard to obtain or non-existent. Scheiber et al. (2023) were able to set up an urban flood model in HCMC based solely on open-access data. However, there are intrinsic uncertainty and limitations introduced by this type of data in such a model and

thus, their main goal was to provide an estimation of preliminary flood maps for the city rather than deterministic conclusions. Our paper seeks to add a case study based on observations that can help validate future statistical and physical modelling efforts in HCMC and address the growing demand for understanding the combined impacts of different flood drivers in the region. In order to do so, high-resolution in-situ measurements of water levels both at the coast and in the river system were gathered and jointly analyzed during an unprecedented extreme event. For these data we applied the harmonic tidal analysis methodology to remove the effect of tides on the water level signal and discern the TY induced surge both at the coast and also, for the first time, upstream in the tidal river. Additionally, we apply the skewness of surge to determine which flood driver dominates. Furthermore, a modified Manning-Strickler law was used together with the analyzed water level signals to estimate river discharge. Finally, we gather in-situ precipitation data and study the performance of several gridded rainfall datasets in estimating the precipitation patterns during TY Usagi. Reported flooding depths and locations are then discussed taking into account land cover and terrain elevation of the region.

The objectives of this paper are i. to provide an observation-based, multi-approach methodology to characterise the drivers of compound flooding after an historical TY, ii. to better understand which of the potential contributors to urban flooding (rainfall-runoff, storm surge or river flood) were most relevant during this particular event and iii. to characterize how the different parts of the hydrological system (terrain elevation, land cover, precipitation, tidal river and coastal surge) contribute to the response of the hydrosystem.

In Sect. 2, we introduce the case study by describing and characterizing the study region, HCMC and the TY Usagi. Then, Sect. 3 presents the methodologies and data that were used to characterize and quantify the impacts. Results are presented in Sect. 4 and discussed in Sect. 5 and followed by a conclusion in Sect. 6.

## 2 Case Study: Ho Chi Minh City and TY Usagi

HCMC is located in southern Vietnam (Fig. 1) in a LECZ where 65 % of its territory is at an altitude below 1.5 m above mean sea level (amsl) (Vachaud et al., 2019). It is characterized by a subtropical monsoon climate with two seasons. The rainy season extends from May to October with a heavy rain period from September to October. The average yearly rainfall over HCMC is 2000 mm of which about 90 % is received during the wet season (Couasnon et al., 2022; Binh et al., 2019; Nguyen et al., 2019a). For a given heavy rain event, the precipitation shows a high variability at short spatial scales (in the order of the km) with considerable differences from one district to another (Vachaud et al., 2019). Urban flooding due to downpours has a high impact on this megalopolis paralyzing it for several hours (Duy et al., 2017) and can already occur for events starting at 50 mm of accumulated rainfall (Vachaud et al., 2019). Together with heavy precipitation, HCMC is also affected by the combined effects of semi-diurnal tidal waves, with amplitude reaching a maximum of 4 m (Schwarzer et al., 2016), and high river discharge from the Saigon river (between -1500 $m^3s^{-1}$ and 2000 $m^3s^{-1}$ (Camenen et al., 2021), where negative discharge represents river flow towards upstream). Tidal fluctuations dominate the totality of the river Saigon from the confluence with the Dongnai river to the outlet of the Dau Tieng reservoir. Thus, affecting both its water levels and discharge, with regular flooding in low-lying urban districts during high spring tides. This flood risk is further exacerbated due to ground subsidence

and the possible impacts of upstream dams on sediment trapping (Marchesiello et al., 2019). The Saigon river is a complex river system, subject to several human and environmental interactions before flowing into the Dong Nai River and finally, into coastal waters. The Saigon river flows from its source in Cambodia to the Dau Tieng Reservoir (270 km$^2$ and 1580·10$^6$ m$^3$) before passing through the HCMC megalopolis. In total, it is 225 km long and its catchment area has a surface of about 4800 km$^2$ (Nguyen et al., 2019b).

The watershed is divided into four parts: the upper part (2538 km$^2$) around the Dau Tieng reservoir; the middle part (1840 km$^2$) which extends south from the reservoir, encompasses the first 80 km of the Saigon and crosses it about 2 km north of the Phu Cuong measurement station; the lower part (548 km$^2$) extending down to the urban center and that encompasses 14 km of river and a complex canal network that flows into it; and, finally, the urban part (211 km$^2$) which encompasses the last, very sinuous 42 km of river, the urban canal network and the Thao Dien measurement station. In the following, we assume that the upper part of the watershed has no direct influence on the river Saigon.

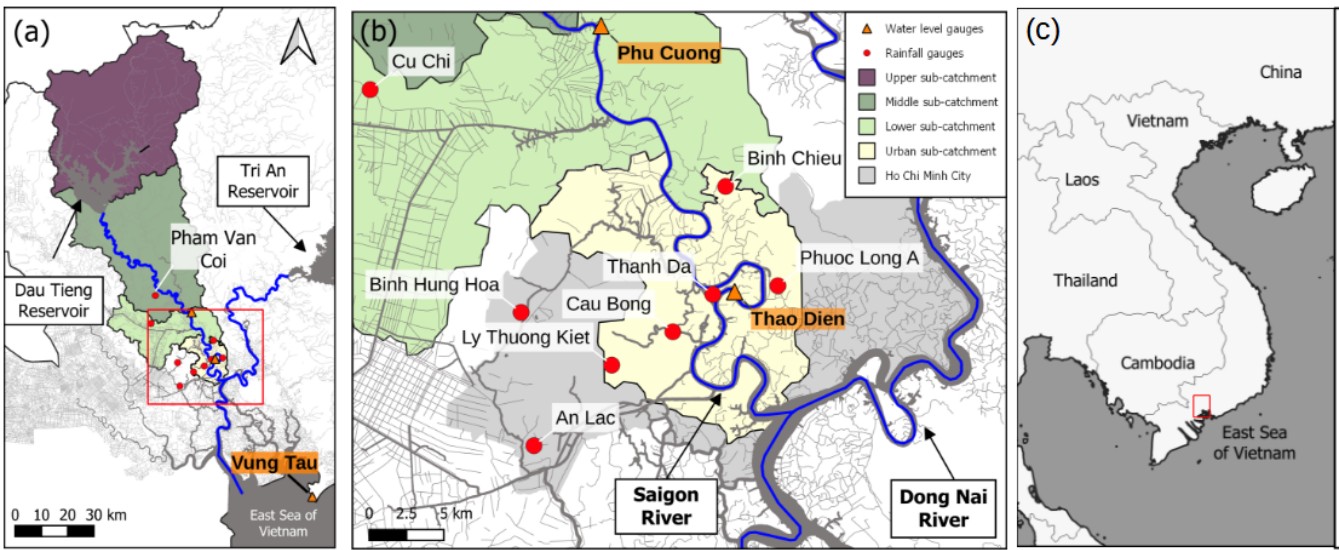

**Figure 1.** (a) The estuary of the Saigon-Dong Nai river system (blue). Grey lines represent the complex, natural and artificial canal network of the area. The Saigon river watershed is divided into four sub-catchments: upper, middle, lower and urban. (b) The Ho Chi Minh City center (grey) and the location of surrounding gauges. The water level gauges (orange triangles) were used for the harmonic tidal analysis and the rainfall gauges (red circles) were used for validation of gridded precipitation datasets. The Pham Van Coi rainfall gauge and the Vung Tau gauge are located outside the spatial extent of (b). The area shown in (b) is represented by the red box in (a). (c) Map of Southeast Asia where the area in red is represented by the red box.

The Saigon River estuary where HCMC is situated borders the East Sea of Vietnam (also known as South China Sea) which is part of the Northwest Pacific Ocean, one of the most active TY basins in the world with about 30 % of the world's annual tropical storms occurring in this region (Trinh et al., 2020). On average, about 4-6 TYs affect the Vietnam coast (Thuy, 2003; Thuy et al., 2016) annually. The expected increase of the number and intensity of TYs due to climate change (Lin et al., 2012)

combined with the low elevation and high population density of HCMC, makes it one of the most vulnerable cities to be impacted by TYs. In addition, the heavy rains and extreme high-water levels associated with TYs aggravate the already high flood risk in the city. In this paper, we focus on one of the most severe rainfall events that this city has ever experienced. It occurred in November 2018, when TY Usagi hit HCMC. During this event, there was more than 300 mm in rainfall over 24 h which led to flooding and considerable material damages.

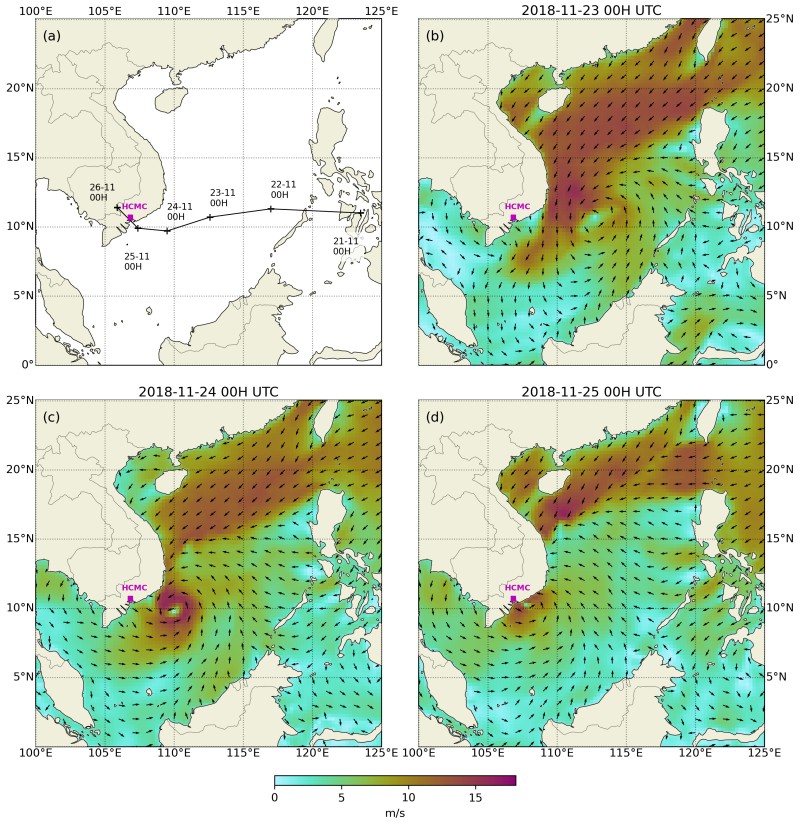

**Figure 2.** (a) The track of TY Usagi and Ho Chi Minh City (pink); (b–d) ERA5 10-m wind fields from November 23rd, 2018 00H UTC to November 25th, 2018 00H UTC.

TY Usagi was a tropical cyclone that affected the Philippines and southern Vietnam in late November 2018, causing severe damage around the Visayas region (Philippines) and HCMC. The storm formed from a disturbance in the Central Pacific basin on November 3rd and developed into a tropical storm (TS) three weeks later, on November 13th. Usagi underwent rapid intensification and peaked in intensity before making its final landfall on the coastal city of Vung Tau just south of HCMC as a weakening tropical storm on November 25th. In Fig. 2 the track of TY Usagi near HCMC is shown using the lowest pressure as indicator. The Joint Typhoon Warning Center (JTWC) assessed its intensity to be equivalent to Category 2 status on the Saffir–Simpson scale. On November 25th, the JTWC downgraded Usagi from TY to a tropical storm as central convection

weakened. Usagi made landfall on Vung Tau, Vietnam at 07:00 UTC as a tropical storm, with the JTWC downgrading Usagi to a tropical depression later that day.

## 3 Methods and Data

The impact of a TY on compound floods in an estuarine system can be separated into inland and coastal drivers (Fig. 3). The first in the form of rainfall-runoff, infiltration and exfiltration influenced by topography and land use and the second in the form of storm surge.

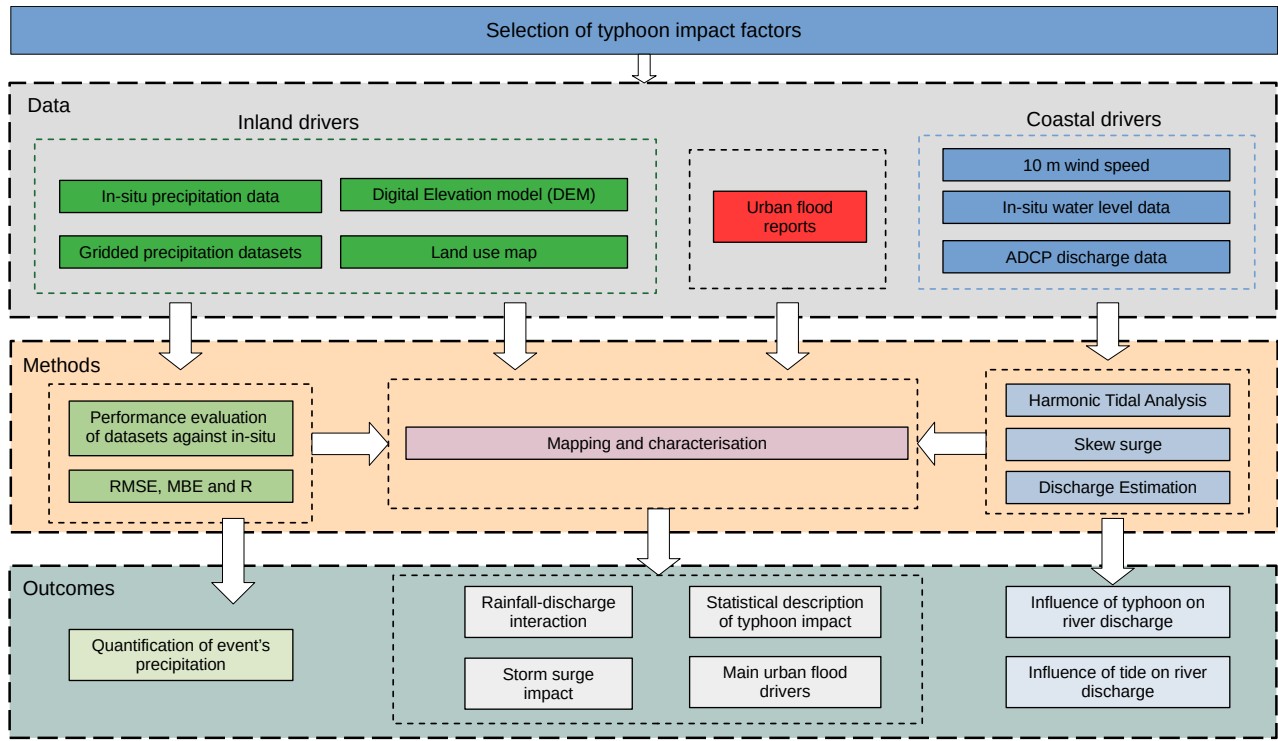

**Figure 3.** Framework to characterize and study the compound flood drivers brought by TY Usagi on the estuarine system. All typhoon-related phenomena that influences coastal dynamics (such as storm surge and wind) is referred to as "coastal" effects. Typhoon-related phenomena influencing the hydrology (precipitation, surface run-off, river discharge, flooding) is referred as "inland' effects.

In order to characterize the impact of TY Usagi on the tidal river system, the following general steps are applied (Fig. 3) and explained in detail in the following sections. We start by selecting a collection of data representative of the factors influencing the impact of a TY. Different gridded precipitation products are evaluated against in-situ measurements to enable a high temporal and spatial resolution study of the precipitation pattern during Usagi. We study topography and land use maps together with the obtained precipitation time series. Additionally, urban flood reports are mapped and analysed in order to form a holistic overview of TY-induced compound flooding before, during and after the event. We then process in-situ water level

measurements using harmonic analysis to obtain the astronomical tide and the residual water level. Furthermore, the skew surge is computed in order to quantify the storm surge and river surge effect on flooding. Then, these signals are used to estimate the discharge of the river Saigon throughout the event. Lastly, the mapping and characterization step encompasses the analysis of the different sources of information and results in order to characterize the impact of this TY on compound flooding in HCMC and upstream of HCMC and clarify the interaction between the different drivers.

## 3.1  Performance evaluation of gridded datasets of precipitation over HCMC

The accuracy of 5 gridded satellite precipitation datasets was evaluated against in-situ measurements. In order to match the location of observations and dataset pixels, the precipitation values of the datasets were spatially interpolated to the location of ground observations using the bi-linear interpolation method (Gaborit et al., 2013; Ang et al., 2022). Additionally, we match the temporal resolution of the datasets and the observations such that a evaluation can be made.

Given the time frame of the heavy precipitation event which started on November, 24th and ended on November 26th we chose the cumulative precipitation over 3 days as the adequate time resolution for the period of available observed precipitation data (2016-2018). Additionally, using the 3-days total instead of a daily total mitigates some daily gaps in the in-situ data. The evaluation was performed over 9 rain gauge locations chosen based on the availability of the data. Some of the rain gauges presented data gaps of several weeks and were not used for the study. Only rain gauges with more than 450 daily measurements

during the selected period (about 3 years) were used. This amount of measurements was chosen given that during the six months of dry season HCMC experiences very few rainy days.

The evaluation was carried out using three statistical indicators: mean bias error (MBE), root mean square error (RMSE) and correlation coefficient (R). These indicators have been widely used in literature for this purpose (Xiang et al., 2021; Ang et al., 2022). MBE is a statistical assessment of the mean difference between the interpolated and observed rainfall, and RMSE

represents the standard deviation between the dataset and the observations. The lower absolute values of MBE and RMSE indicate the better performance of the gridded dataset. R measures the degree of linear correlation between the gridded and observed datasets. Higher values of R indicate higher accuracy of the product in estimating precipitation. The equation, range, and optimal value of each index are presented in Table 1.

**Table 1.** Equations and optimal values of statistical indices. $P_i$ and $O_i$ denote predicted and observed values, respectively, of precipitation on the $i^{th}$ day. $cov(P,O)$ denotes the covariance between the predicted and observed values.

| Indices | Equation | Range | Optimal values |
|---------|----------|-------|----------------|
| Mean Bias Error | $MBE = \frac{1}{N}\sum_{i=1}^{N}(P_i - O_i)$ | $-\infty$ to $\infty$ | 0 |
| Root Mean Square Error | $RMSE = \sqrt{\frac{1}{N}\sum_{i=1}^{N}(P_i - O_i)^2}$ | 0 to $\infty$ | 0 |
| Correlation Coefficient | $R = \frac{cov(P,O)}{s_P s_O}$ | -1 to 1 | 1 |

**Table 2.** Overview of the precipitation data used in this study. The categories are indicated as follows: gauge-based data (G), satellite-based data (S), reanalysis data (R) and in-situ data.

| Name | Variable | Category | Temporal/Spatial Resolution | Temporal Coverage | Reference |
|------|----------|----------|------------------------------|-------------------|-----------|
| GPCC | Rainfall | G | Daily/1.0° | 1982-2019 | Ziese et al. (2020) |
| CMORPH v1.0 | Rainfall | S | 3-hourly/0.25° | 1998-2021 | Joyce et al. (2004) |
| MSWEP v2.0 | Rainfall | G, S, R | 3-hourly/0.25° | 1979-2021 | Beck et al. (2019b) |
| ERA5 | Rainfall | R | Hourly/0.5° | 1979-2021 | Munoz-Sabater et al. (2021) |
| IMERG | Rainfall | S | 3-hourly/0.1° | 2000-2021 | Huffman et al. (2019) |
| Urban rain gauges | Rainfall | in-situ | daily/N.A. | 2016-2018 | HCMUDC - HOS |

## 3.2 Ground-based and satellite precipitation datasets

Table 2 summarizes the precipitation datasets and observations used in this study. Gridded precipitation products can be classified into three categories based on differences in the data sources and retrieval models including gauge-interpolated, satellite-based, and reanalysis precipitation datasets (Jiang et al., 2021). Satellite-based datasets use polar-orbiting passive microwave (PMW) sensors on low-Earth-orbiting satellites and geosynchronous infrared (IR) sensors on geostationary satellites to estimate precipitation (Larson and Peck, 1974; Kidd and Huffman, 2011). Often this datasets offset their limited abilities by blending rain gauge data with their measurements (Mizukami and Smith, 2012). Reanalysis-related datasets generate several meteorological variables with a consistent spatial and temporal resolution by assimilating observations such as weather stations, satellites, ships, and buoys based on different climate models. In this analysis five datasets that belong to the four categories were chosen in order to examine their different performances over HCMC. All datasets that were used are freely available online via public repositories.

*Global Precipitation Climatology Centre (GPCC) dataset.* The GPCC provides the largest gauge-based dataset product derived from quality controlled station data sourced from national meteorological and hydrological services, global and regional data collections as well as WMO GTS-data. This GPCC product is recommended to be used when the daily precipitation information is of highest importance, e.g. for analyses of extreme events and related statistics at daily resolution. Despite its low spatial resolution, the GPCC product was chosen as the gauge-based dataset since it is recommended for analyses of extreme events and related statistics at daily resolution (Jiang et al., 2021).

*Climate Prediction Center MORPHing technique (CMORPH) dataset.* This product consists of satellite precipitation estimates that have been bias corrected and reprocessed using the CMORPH to form a global, high resolution precipitation analysis at very high spatial and temporal resolution. We chose this product due to its advection scheme of cloud features and wide range

of applications including hydrological studies and extreme event analysis (Joyce et al., 2004).

***Multi-Source Weighted-Ensemble Precipitation (MSWEP).*** MSWEP is a global precipitation product which incorporates daily gauge observations and accounts for gauge reporting times to reduce temporal mismatches between satellite-reanalysis estimates and gauge observations. It was chosen for the study because it merges gauge, satellite, and reanalysis data to obtain precipitation estimates. MSWEP has demonstrated higher overall accuracy than other widely used precipitation products in both densely-gauged and ungauged regions (Beck et al., 2017, 2019a). Beck et al. (2019b) provides a detailed description of the MSWEP V2.2 methodology.

***European Centre for Medium-Range Weather Forecasts (ECMWF) Re-Analysis product (ERA5).*** The ERA5 product is the fifth-generation atmospheric reanalysis to replace ERA-Interim produced by ECMWF. It assimilates observations from over 200 satellite instruments or types of conventional data and information on rain rate from ground-based radar-gauge composite observations. Even though ERA5 has shown stronger data deviations than in bias-corrected satellite-based precipitation datasets (Islam and Cartwright, 2020), it was chosen since it can capture the spatial distribution of TY precipitation centers (Jiang et al., 2021).

***Integrated Multi-satellitE Retrievals for Global precipitation measurement (IMERG).*** The IMERG product combines information from the GPM satellite constellation to estimate precipitation over the majority of the Earth's surface. The product fuses the early precipitation estimates collected during the operation of the TRMM satellite (2000 - 2015) with more recent precipitation estimates collected during operation of the GPM satellite (2014 - present). IMERG was chosen given that it generally shows high accuracy and good performance in hydrological simulations when compared with the ground observations (Ang et al., 2022) as well as its good capability in extreme rainfall events applications (Huang et al., 2019).

***Rain Gauge observations.*** We use daily rainfall data from nine rain gauges located in and around the city center of HCMC as shown in Figures 1a) and 1b). This data was provided by the Ho Chi Minh Urban Drainage Company (HCMUDC) and the Hydrometeorological Observatory - Southern region (HOS) for the period from May 1st, 2016 to November 30th, 2018.

### 3.3 Harmonic Tidal Analysis

The Saigon river discharge is highly influenced by the mixed, semi-diurnal tidal cycle and presents a relatively low net discharge (Nguyen et al., 2019b, 2020; Camenen et al., 2021) which is one order of magnitude lower than the instantaneous discharge. The water level time series are highly influenced by the tidal signal making it difficult to access the impact of TY Usagi on both river and coastal water levels. It is thus important to process and filter out the tidal component from observed time series. According to Cid et al. (2017), we first remove the average water level variability from the water level time series by subtracting the monthly moving average. Then, we follow the methodology proposed by Pugh and Woodworth (2012) to produce the tidal signal via classical harmonic analysis to extract information on the amplitude and phase of the tidal constituents. Classical

harmonic analysis has been developed and widely used to analyze and predict tides (Jin et al., 2018; Trinh et al., 2020; Couasnon et al., 2022). The UTide package developed by Codiga (2011) and implemented in Python by Bowman (2020) was used for extracting the tidal constituents. This package provides harmonic analysis with up to 146 tidal constituents. We use all available constituents for our tidal prediction. However, we exclude the constituents whose harmonic constants usually include mostly quasi-periodic meteorological effects (Mm, MSf, Mf, Sa, Ssa, S1). Excluding these long-term constituents from the analysis avoids over fitting the tidal prediction with frequencies that capture non-astronomical effects on water level such as wind, temperature and atmospheric pressure (Parker, 2007). Additionally, the upstream location of the river sensors justifies the use of constituents produced by nonlinear mechanisms in shallow water. These mechanisms alter the characteristics of the tidal waves and may include the effect of bottom friction, standing wave generation or local resonances due to interaction with varying topography (Pugh, 2004). Hence, we use a total amount of 140 constituents.

In order to obtain the non-tidal effects on water level we compute the difference between the observed water level and the predicted tide, i.e. the residual signal.

### 3.4 Skew Surge

The residual signal at the sea level gauge of Vung Tau (Fig. 1a) presents sporadic peaks and troughs due to small shifts in tidal phase. This is a common challenge in tide prediction analysis in systems with large tidal amplitudes which is the case here (Williams et al., 2016; Calafat and Marcos, 2020; Couasnon et al., 2022). Hence, in addition to the residual signal, we look at the skew surge which represents the excess water level over the high tide. This metric has seen relevant application in several coastal flooding studies (Williams et al., 2016; Haigh et al., 2016; Couasnon et al., 2022).

### 3.5 Water level data

Time series data at 10 minute intervals were obtained from a 2-year measurement campaign (2017-2018) directed by the Centre Asiatique de Recherche sur l'Eau (CARE). Measurements were performed using CTD Diver sensors at two locations: Phu Cuong and Thao Dien as shown in Figure 1. Phu Cuong and Thao Dien are located at about 137 and 60 km from the coast (along river), respectively. In Camenen et al. (2021) the quality of these measurements has been validated against data of the Center of Environmental Monitoring (CEM) of Vietnam. This data is complemented by the record at the Vung Tau tide gauge from the research quality dataset available through the Joint Archive for Sea Level of the NOAA National Centers for Environmental Information (Caldwell et al., 2015). A statistical description of these datasets can be found on Table C1 in Appendix C. The stations of Phu Cuong, Thao Dien and Vung Tau depicted in orange in Fig. 1 will also be referred to as the upstream, urban and coastal stations, respectively.

## 3.6 River Discharge Estimation

The instantaneous river discharge was estimated by applying a stage-fall-discharge (SFD) rating curve adapted from the general
Manning-Strickler law (Eq. 1), previously tested and validated by Camenen et al. (2017) and used to predict the total discharge
of the Saigon river in Camenen et al. (2021):

$$Q(t) = sign(S) \cdot K \cdot A_w(t) \cdot R_h(t)^{2/3} \cdot \sqrt{|S(t+dt)|}, \tag{1}$$

with $Q$ the river discharge $[m^3 s^{-1}]$, $K = 27.06 \ m^{1/3} s^{-1}$ the Manning-Strickler coefficient , $R_h = A_w/P_w$ the hydraulic
radius $[m]$, $A_w$ the wet section $[m^2]$, $P_w$ the wet perimeter $[m]$. Note that $A_w$ and $P_w$ are both a function of the water level
and thus, of time. The term $sign(S)$ is equal to the sign of the slope, $S$, taking the values of +1 or -1. The energy slope, $S$ [-],
is assumed equal to the water slope and is computed as:

$$S = \frac{H_{up} - H_{dn} + dH}{L}, \tag{2}$$

with the water level $H_{up}$ and $H_{dn}$ measured at Phu Cuong (PC) and Thao Dien (TD), respectively, and $L$ the distance
between the two locations. We use the term 'water level ($H$)' to refer to the height of the column of water above the pressure
gauges in the river. The tidal oscillations are propagated from the coastal tide gauge to the river gauges. Since there is no fixed
datum between river and tide gauges, we normalize all signals by mean removal. This makes the tidal harmonics to oscillate
about zero for all gauge locations thus, making them comparable. In addition, the dH parameter in the equation 2 provides
an additional calibration parameter that helps mitigate this problem. The term dt is a time lag required to account for the
propagation of the tidal wave between one location to the other. The full observed water levels at Phu Cuong and Thao Dien
were used to compute the total discharge of the Saigon river. The predicted tidal signals obtained via harmonic tidal analysis at
these stations were used to compute the discharge due solely to tidal fluctuations. Then, the tidal discharge is subtracted from
the total discharge in order to obtain a residual discharge - the discharge due to non-tidal effects.

In Camenen et al. (2021), the model calibration is done using two ADCP campaigns: i. March 2017 during an asymmetric
tide and ii. September 2016 during a symmetric tide. During the asymmetric tide the equation has much more difficulty
following the discharge measurements than during the symmetric tide. The parameters to be calibrated are the following:

- K, the Manning-Strickler coefficient of the river reach is a measure of channel roughness or friction and is assumed
  constant.

- dt, is a time lag required to account for the propagation of the tidal wave between the downstream location to the upstream
  location.

- dH, is used to compensate for the fact that the reference points of each location are different and unknown.

The parameters K, dt and dH are calibrated one at a time to optimize the Root Mean Square Error (RMSE) which provided good results. However, in this study we improve this calibration by using a non-linear least squares fitting technique (not presented here). The optimal parameter values found for this study were dH = -0.149 m and dt = -2 h. This calibration method yielded better results for the estimation of discharge than in Camenen et al. (2021). We improve the RMSE of total discharge during an asymmetric tide from 350 $m^3s^{-1}$ to 185 $m^3s^{-1}$ using K = 27 $m^{1/3}s^{-1}$, dt = 2 h and dH = -0.15 m. Additionally, statistical information about the estimated discharge can be found in Table C3 in Appendix C.

## 3.7 Other data

*Topography.* The topography maps were obtained from the NASA Shuttle Radar Topographic Mission (SRTM) 90 m DEM Digital Elevation Database. The SRTM mission has provided digital elevation data (DEMs) for over 80 % of the globe. This data is currently distributed free of charge by USGS and is available for download from the National Map Seamless Data Distribution System, or the USGS ftp site. The vertical error of the DEM's is reported to be less than 16m (Farr et al., 2007). The SRTM vertical datum is global mean sea level and is based on the WGS84 Earth Gravitational Model (EGM 96) geoid (EROS, 2018). Throughout the manuscript when using the term "mean sea level" we refer to the global mean sea level used as datum for the SRTM data.

*Land Use.* The land use map of the region around HCMC was obtained from the Large-scale Land Use Land Cover (LULC) website (https://www.eorc.jaxa.jp/ALOS/en/dataset/lulc/lulc_vnm_v2109_e.htm). LULC information was derived using a random-forest-based algorithm and several geospatial data sources, including Landsat and Sentinel-1 and -2 imagery (Phan et al., 2021). The final product contains annual land cover information from 1990 to 2020 in Vietnam at a 30 m resolution. This data is independently validated with field surveys and visual interpretation data. The overall accuracy of the level-1 layer ranges from 86 % to 92%.

*Wind.* Storm surge is produced by water being pushed onshore by the force of cyclonic winds. The impact on surge of the low pressure associated with intense storms is minimal in comparison to the water being forced toward the shore by the wind (NOAA, 2023). Additionally, many other factors, such as angle of approach of the TY, radius of maximum winds and the slope of the continental shelf may also have an influence (Sebastian et al., 2019). Storm size also significantly contributes to the generation of storm surge (Trinh et al., 2020) and provides an indication of the spatial region influenced by the TY. Larger TYs create higher storm surges and coastal inundation (Orton et al., 2015). Therefore, we use the wind field to determine wind direction and the size of the TY Usagi (Fig. 2). ERA5 outputs for 10 m u and v wind components were used to map the approach of TY Usagi towards the southern coast of Vietnam. This data is provided free of charge at the Copernicus Climate Data Store (https://cds.climate.copernicus.eu) with an hourly resolution and global grid of 0.25° for the period 1959 to present (Hersbach et al., 2020).

*Flood data.* Urban flood reports for November 25th, 2018 were provide by the DECIDER project (https://www.decider-project.org) and originally obtained from the former Steering Center for Flood Control (SCFC) in HCMC. These include coordinates of flooded streets and corresponding flood height. Additionally, a flood map based on manufacturing firms' reports was obtained from Leitold et al. (2021).

**Table 3.** Overview of water level, flood reports, topography and land use data used in this study. The categories are indicated as follows: gauge-based data (G), satellite-based data (S), reanalysis data (R) and in-situ data.

| Name | Variable | Category | Temporal/Spatial Resolution | Temporal Coverage | Reference |
|------|----------|----------|------------------------------|-------------------|-----------|
| ERA5 | 10 m wind | R | hourly/0.25° | 22-27/11/2018 | Hersbach et al. (2020) |
| Tide gauge | Sea level | in-situ | hourly/N.A. | 2007-2021 | Caldwell et al. (2015) |
| Urban river gauge | Water level | in-situ | 10 min/N.A. | 2017-2018 | Camenen et al. (2021) |
| Flood reports | Location and flood depth | report | N.A./N.A. | 25/11/2018 | SCFC |
| Flood reports 2 | Location | report | N.A./N.A. | 25/11/2018 | Leitold et al. (2021) |
| SRTM DEM | Elevation | S | N.A./90 m | N.A. | Farr et al. (2007) |
| Land Use | Land use | S | Yearly/30 m | 2018 | Phan et al. (2021) |

## 4 Results

### 4.1 Assessment of precipitation products

The results of all statistical indices (RMSE, MBE and R) for each station are presented in Fig. 4. The worst performing dataset over this domain is ERA5 with large values of RMSE and MBE (both > 100 mm/3 days) and low linear correlation (R ≤ 0.4). This indicates that ERA5 is overestimating rainfall over the whole domain. The ERA5 dataset that was evaluated has a 0.5° grid which is not capable of capturing the precipitation patterns over HCMC. The other gridded products perform relatively

well in stations within the city center (20 ≤ RMSE < 50; -20 ≤ MBE < 50 mm/3 days). This indicates that all datasets, with the exception of ERA5, are able to estimate rainfall patterns over HCMC. The gauge-based dataset GPCC is able to do so with the most coerce grid: a daily temporal window and spatial resolution of 1°. Even though CMORPH presents a sophisticated advection scheme (0.25°), its performance is similar to the higher spatial resolution IMERG (0.1°) across all metrics. Additionally, the performance of MSWEP (0.25°) is slightly better than CMORPH over the city center. However, this

better performance can also be attributed to the incorporation of in-situ data in addition to the satellite data used to produce MSWEP. Nonetheless, this shows that a grid size of 0.25° suffices to estimate rainfall over HCMC using hybrid measurements sourced from satellites and in-situ data.

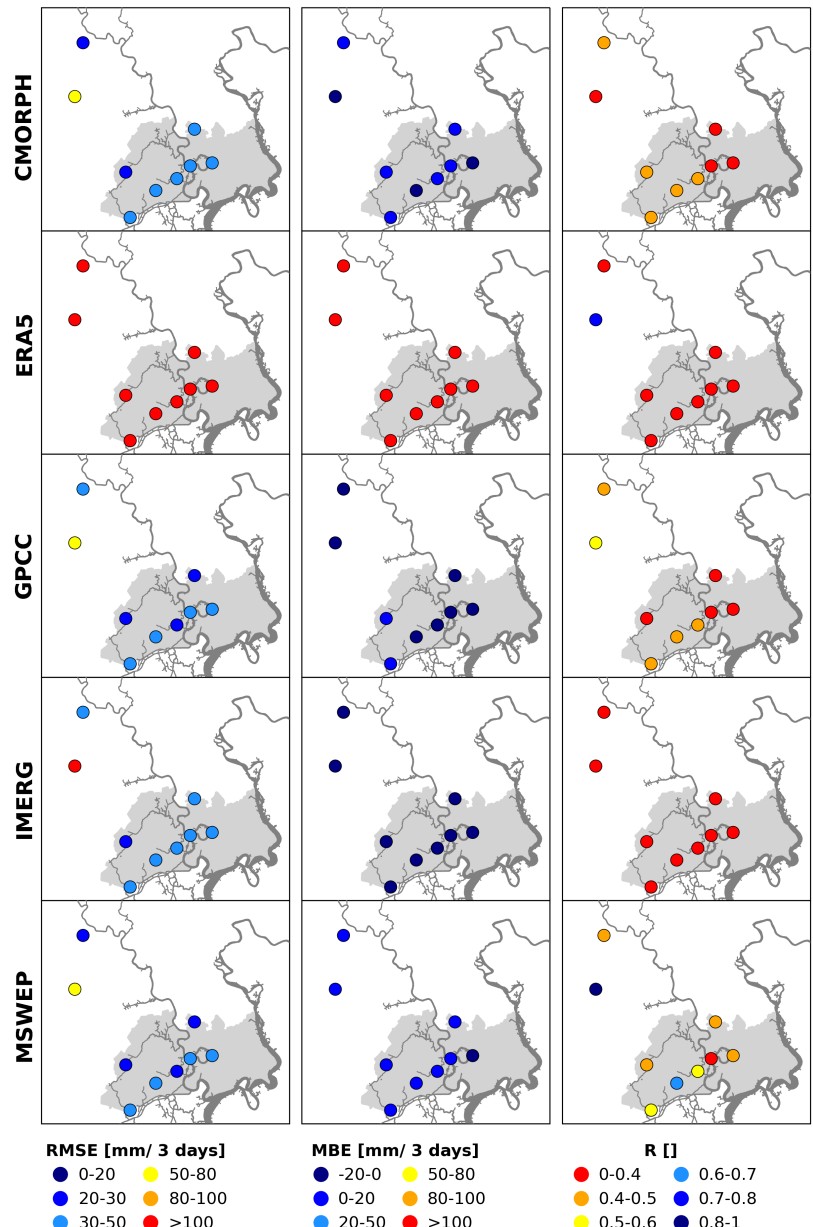

**Figure 4.** Spatial distribution of root mean square error (left), mean bias error (middle) and correlation coefficient (right) between 5 gridded products and gauged observations at daily time scale during 2015-2018. The circles denote rainfall gauging stations located in and around the Ho Chi Minh City area (light grey). The Saigon-Dong Nai system is represented in dark grey.

The degree of linear correlation between all datasets and rainfall gauges is always below 0.7 for almost all stations. MSWEP shows higher values of R ($0.4 \leq R < 0.7$) than all other datasets. In fact, all other datasets show very low values of R (R < 0.5) indicating that these datasets have a weak linear relationship with the observed daily data. The main factor contributing

to the difficulty in accurately portraying the precipitation over HCMC is the convective pattern of precipitation events leading to the non-uniform distribution of rain at the scale of the city (Vachaud et al., 2019). It can happen that it rains heavily in one district whereas in another it does not rain at all. The spatial resolution of all datasets (always at least $\geq 0.1°$) remains too coarse to accurately capture the spatial variability during rainfall events. Additionally, this evaluation is undertaken at the relatively high timescale of the day against scarce insitu measurements (both in time and space) which impacts the performance of the datasets.

Overall, MSWEP is the dataset that better represents the precipitation patterns over HCMC by either showing similar metrics or outperforming other datasets especially in coefficient of correlation, R. In order to further validate this dataset, the performance metrics (RMSE, MBE and R) were computed for the moth of November 2018 on a daily timestep. In Fig. 5 the precipitation products are plotted against the available rain gauge data (in grey bars). The extreme rainfall brought by TY Usagi on November 25th, 2018 is clearly seen in both rainfall gauge and datasets. For this period the rain gauge data is scarce with 3 out of 9 stations not having any measurements namely, Pham Van Coi, Thanh Da and Cau Bong. Additionally, the stations that present measurements have important data gaps. All datasets to the exception of ERA5 underestimate rainfall during Usagi. ERA5 (brown curve in Fig. 5) overestimates it with values above 500 mm for the day that Usagi made landfall. However, for the presented rainfall gauge measurements MSWEP (in light blue) seems to capture the precipitation behaviour during TY Usagi better than all other datasets (Table B1 in Appendix B). For MSWEP, the average statistics over the month of November 2018 and over all stations for RMSE, MBE and R are 7.6 mm/day, -4.3 mm/day and 0.6, respectively.

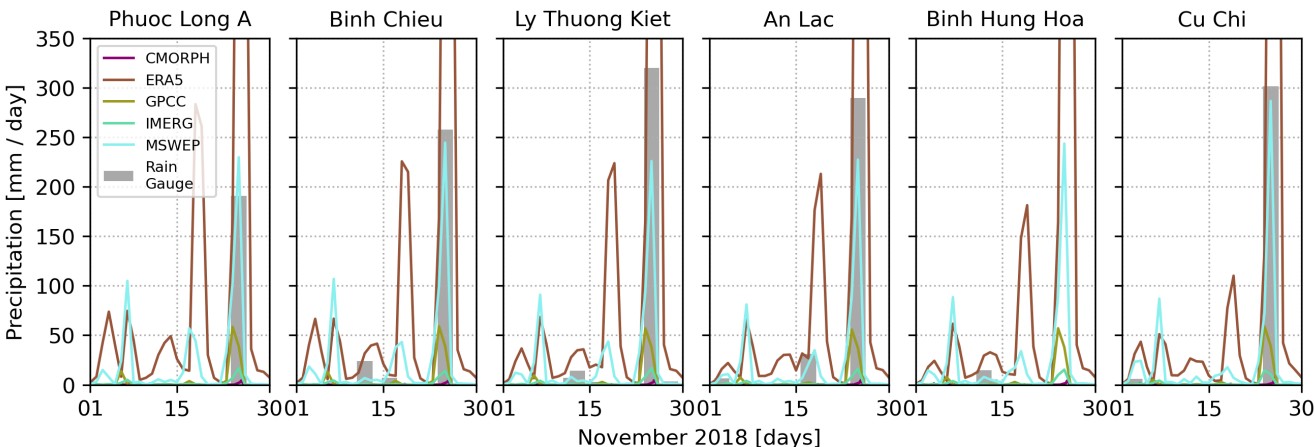

**Figure 5.** The 5 precipitation products compared to the 6 rain gauges with available observations for the month of November 2018. The peak of precipitation during TY Usagi is best captured by MSWEP (light blue) whereas the other datasets either underestimate or overestimate it. The ERA5 (brown) presents estimations above 500 mm. The high spatial variability over the city makes it hard to obtain accurate results at all stations simultaneously.

## 4.2 Statistical analysis of data and characterisation of TY Usagi

In this section we perform a seasonal statistical overview of water level, discharge and precipitation. Additionally, we quantify the effect of TY Usagi on these hydrological variables in comparison to seasonal variations. The boxplots referring to this data are presented in Fig. 6. The x-axis for the water levels are separated into the direct measurement (referred to as "Observed"), the tidal signal predicted via harmonic tidal analysis ("Tidal"), the signal without tidal influence ("Residual") and the computed skew surge ("Skew surge"). Similarly, for the discharge computed at Phu Cuong the estimations are divided into "Observed", "Tidal" and "Residual".

### 4.2.1 Seasonal patterns

**Predicted tide:** As can be see in Figures 6(a), 6(b) and 6(c), the predicted tide signal presents no seasonality as its average, median and interquartile range do not vary between wet and dry season. However, the tidal signal changes from Vung Tau at the coast towards upstream at Phu Cuong in the river Saigon. The whiskers in the boxplots decrease in size from coast to upstream which means tidal amplitude decreases. We verify a decreasing tidal amplitude from 3.9 m (4 m) at the coast, to 3.1 m (3.2 m) at the urban city center, and to 2.7 m (2.5 m) upstream during wet (dry) season. This is equivalent to a decrease in tidal amplitude of 8 mm per along river km between the coast and the city center and of 11 mm per along river km between the city center and upstream of HCMC. It can also be seen that negative water levels can reach greater values in modulus than positive water levels. This is due to the very low low tides during spring tidal cycles, as illustrated in Fig. 7(a). Additionally, the size of the boxes decreases upstream which indicates the data is less spread in the river than at the coast, as the interquartile range (IQR) is smaller. Furthermore, the box is closer to the higher values of water level at all stations. Hence, the probability density function of the tide is skewed towards higher values. This is due to the asymmetric nature of the tide with cycles where low tide is little pronounced and the lowest water level differs by less than 1 m from the neighbouring high tides (see Fig. 7a).

**Observed water levels:** The observed water level signals show average water levels that are higher than the tide prediction during the dry season and lower during the wet season. This behaviour is counter intuitive from an hydrological point of view where water levels are strongly connected to seasonal rains. This seasonality pattern in the coastal water levels is in line with Trinh et al. (2020) and it propagates upstream as shown in Camenen et al. (2021). This clarifies that river water levels are interconnected to seasonal coastal storm surge caused by the wind pattern of the East Asian summer monsoon, typically from November to April (Marchesiello et al., 2020). Thus, the monsoon wind overpasses precipitation effects on the Saigon river.

**Residual water levels and skew surge:** For both seasons we expect the same behaviour as the observed time series and thus, higher average residuals in dry season than in wet season. The difference on average residual between seasons is of 24 cm at the coast, 26 cm at the urban center and 18 cm at the upstream station. As for the seasonal difference of average skew surge we find 19 cm at the coast, 23 cm at the urban center and 17 cm at the upstream station. These results show that at the urban center we have consistently higher surges than at the coast or than upstream which indicates that the surrounding highly urbanized land cover and complex canal system are playing a role in the water levels at this location. Additionally, both residual signals and skew surge take on more negative values in the wet season as all values below the upper quartile (Q3) are negative. The

inverse is seen for the dry season as all values above the first quartile (Q1) are positive. This behaviour is due to the difference in tidal and observed water levels. Observed water levels are lower than predicted in the wet season leading to negative average residuals and skew surge and vice-versa for the dry season. This further justifies a seasonal surge in the river due to a seasonal surge at the coast.

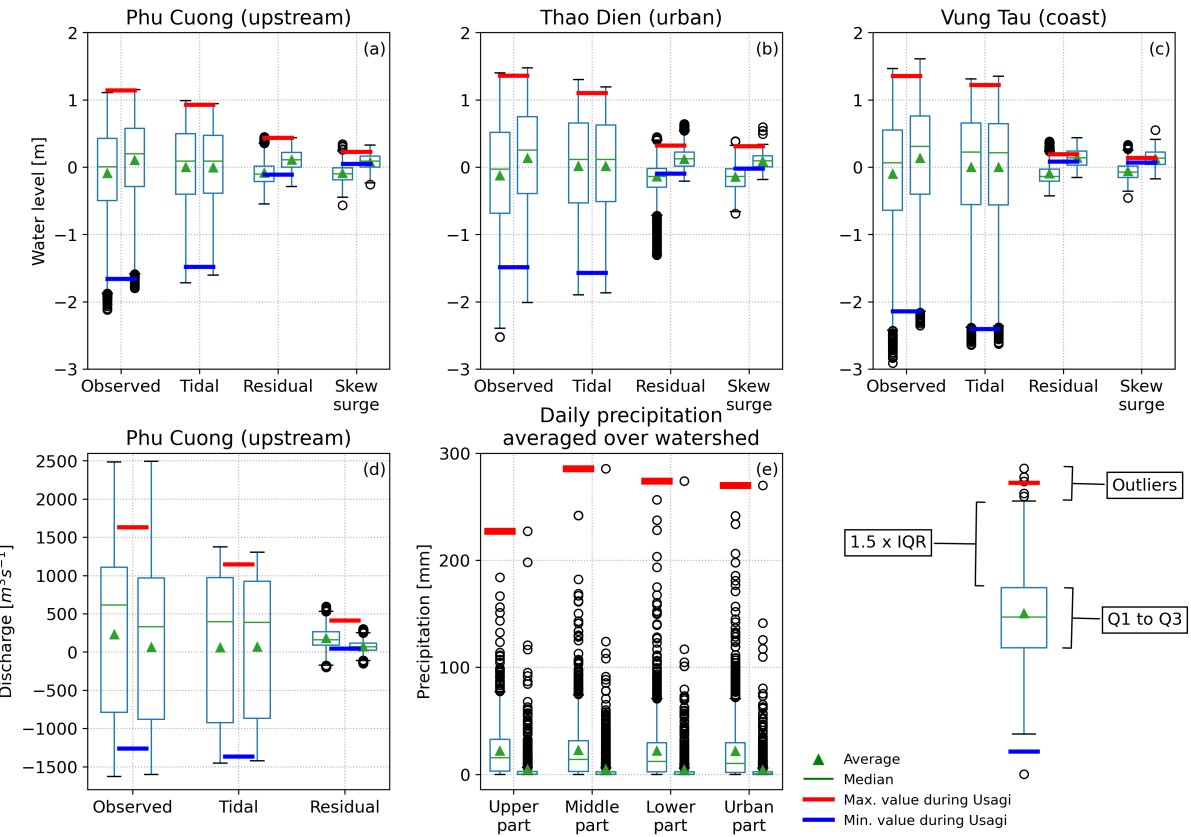

**Figure 6.** Boxplots of water level (observed signal, tidal prediction, residual signal and skew surge) for the upstream station of Phu Cuong (a), the urban station of Thao Dien (b) and the coastal station of Vung Tau (c). Boxplots of estimated discharge (using observed water level signals, tidal prediction and residual signals) for Phu Cuong (d). Boxplots of daily cumulative precipitation averaged over the spatial extent of the upper, middle, lower and urban parts of the watershed (e). Each variable in the x-axis has two boxplots corresponding to wet (left) and dry (right) seasons. A legend of the boxplots is provided in the bottom right corner: the box extends from the first quartile (Q1) to the third quartile (Q3) and the whiskers extend from the box by 1.5 times the interquartile range (IQR = Q3 - Q1). Extreme values for TY Usagi are reported in red (max) and blue (min) horizontal bars and correspond to the time window from 24-11-2018 to 26-11-2018. The time period of the data presented here correspond to the temporal coverage in Tables 2 and 3.

**Discharge:** The estimated discharge boxplots (Fig. 6d) show similar discharge due to tides during both seasons. However, the observed discharge differs from wet to dry season. The dry season tidal discharge is similar to the observed discharge on average, median and IQR. However, the observed discharge signal presents longer whiskers and thus, higher discharge values

with observed discharge between -1604 and 2492 m$^3$s$^{-1}$ versus -1500 and 1500 m$^3$s$^{-1}$ for the tidal discharge. This means that in the dry season tides are the main driver for discharge in the Saigon and less frequent but higher values occur due to other factors. Such factors include the outlier precipitation events during the dry season (see Fig. 6e). On the other hand, the wet season average and median discharge are respectively 168 and 217 m$^3$s$^{-1}$ higher than for the tidal discharge. The larger observed discharge in the wet season is could be attributed to the more intense rainfall during this period: the average wet season daily rainfall is 22 mm against 5 mm in the dry season averaged over the full watershed (Fig. 6e). As a result, the average residual discharge in the wet season is larger and the distribution is more spread (larger IQR) than in the dry season. This translates to an average residual discharge of 180 m$^3$s$^{-1}$ in the wet season and 72 m$^3$s$^{-1}$ in the dry season.

**Rainfall:** In Fig. 6(e), it can be seen a clear increase of rainfall from dry to wet season over the whole watershed as expected. We see a lot of outlier events in both seasons with the dry season sometimes having exceptionally high rainfall events that are well above the average even for the more rainy wet season. The increased wet season rainfall seems to not change the water levels at the seasonal time scale. This is because we have a clear increase in rainfall in the wet season but a decrease in water levels over the two river stations. This shows that coastal water level is the main driver of river water levels and not rainfall over the whole extension of the Saigon river. On the other hand, we see a slight increase in discharge during the wet season.

### 4.2.2 Effect of TY Usagi (24-26 November, 2018)

**Water levels:** From Fig. 6(c), it can be seen that TY Usagi occurred during a large amplitude spring tide (see also Fig. 7) with a low tide below Q1 of the tidal levels at the coast. At this location both residual water level and skew surge are within the IQR for the dry season. Hence, one can infer that the storm surge at the coast was not statistically significant when compared with the distribution of values during this season. Furthermore, the maximum coastal residual water level and skew surge were of 19 cm and 14 cm, respectively, which corresponds to 5 cm and 1 cm above the respective dry season averages. At the urban city center (Fig. 6b) skew surge is close to being an outlier with a value of 31 cm which is 22 cm above the seasonal average. The residual signal maximum would be an outlier in the wet season and in the dry season it is above Q3 with a value of 32 cm which is 20 cm above the seasonal average. Finally, at the upstream station skew surge due to Usagi is less statistically significant than at the city center with a value of 23 cm which is above Q3. However, the residual signal is stronger at this location than at the city center with a value of 44 cm and 33 cm above the seasonal average.

**Discharge:** The Usagi impact seems to have brought about higher discharge at the upstream location of Phu Cuong than usual. The discharge peak sits well above Q3 for the wet season and is a clear outlier for the dry season. The peak discharge during Usagi was 413 m$^3$s$^{-1}$ which corresponds to 233 m$^3$s$^{-1}$ above the average of the wet season. It can also be seen that there have been higher discharges in the wet season during the period of this data even though it is clearly an outlier discharge for the dry season.

**Rainfall:** In Fig. 6e) it is confirmed that the rainfall brought by Usagi constituted the maximum daily rainfall in this data series for both wet and dry seasons. On the day of the landfall (November 25th, 2018), it rained on average 227 mm over the upper part, 285 mm over the middle part, 274 mm over the lower part and 270 mm over the urban part of the watershed (see Fig. 1).

### 4.3 Coastal driver: Storm surge effect on river dynamics

The results from the harmonic tidal analysis are presented in Fig. 7 for the period of one month around the Usagi event. Additionally, a statistical description of the residual water level signal can be found in Table C2 in Appendix C. The predicted tide, shows a mixed semi-diurnal character for the three stations (Fig. 7b). The observed time series of water level (Fig. 7a) illustrates how the tidal wave propagates from the coast at Vung Tau to the upstream river stations. The tidal forcing is dominant along the river system even throughout the Usagi event where no direct impact can be observed on the water level timeseries. This is expected (see Sect. 4.2) and further certifies that the Saigon river's hydrodynamics are mainly controlled by tidal forcing.

A clear anomaly during the event can be seen when using the harmonic tidal analysis methodology to obtain a water level without tidal influence (namely, the residual water level, Fig. 7c). Both river stations present an increase in water level after landfall (Fig. 7c, black and grey lines). On the other hand, the coastal response is not evident (Fig. 7c, blue line). This indicates that TY Usagi had an inland effect on the hydrological system rather than a coastal effect. Additionally, a substantial decrease in residual water level at the two river stations (Thao Dien and Phu Cuong) prior to landfall can be seen (Fig. 2a). This suggests that at the timescale of the event river and coastal water levels are decoupled: river stations experience negative residual prior to landfall and a significant surge after landfall whereas the coastal station does not.

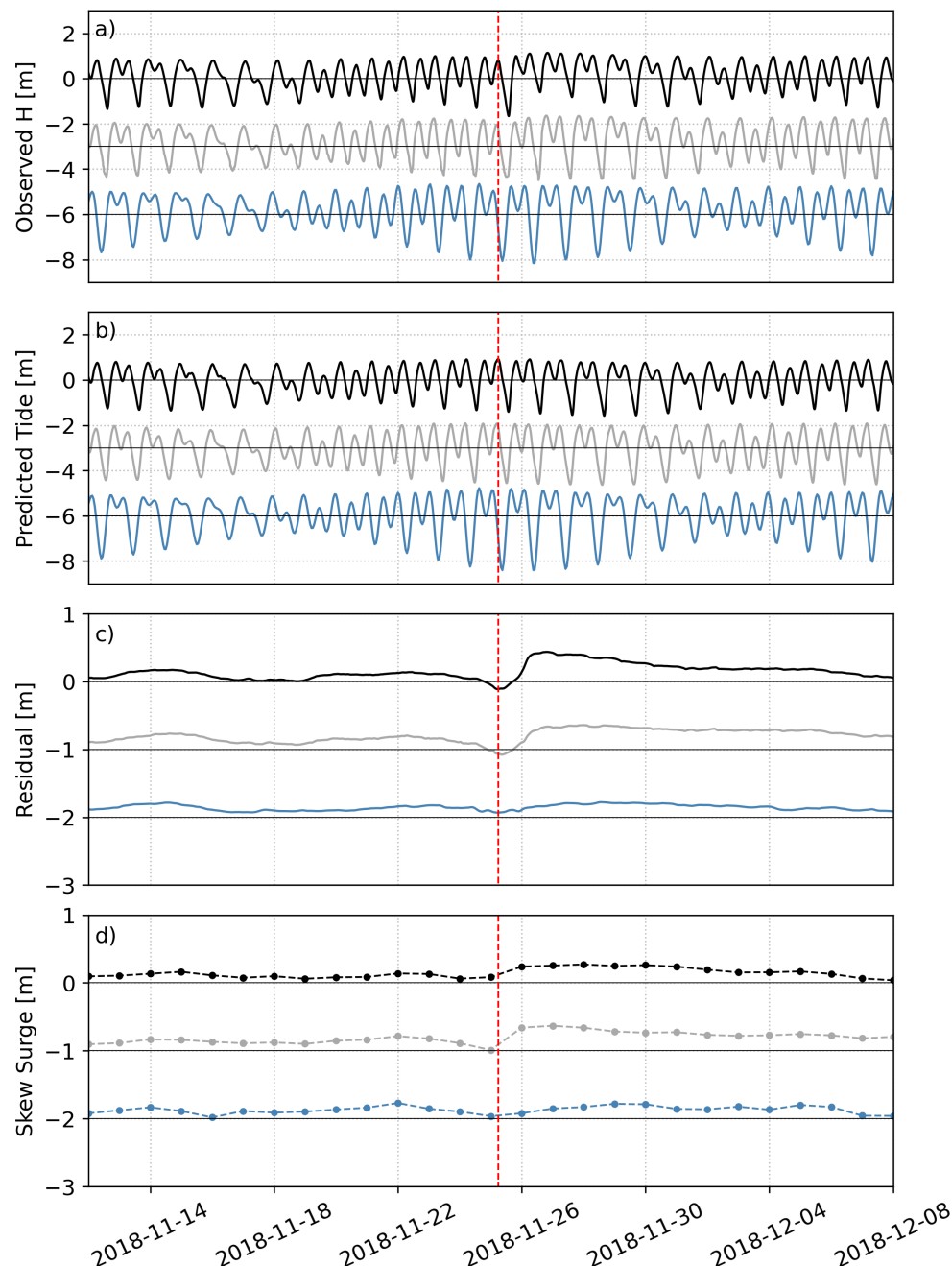

**Figure 7.** Result for tide prediction at the river stations - Phu Cuong (black) and Thao Dien (grey) - and the coastal station Vung Tau (blue). Timeseries of the full observed signal (a), the tidal prediction (b), the residual (c) obtained by subtracting the tidal prediction from the observed signal and the skew surge (d). For the sake of readability, the signals in (a) and (b) are displaced by 3 meters and the signlas (c) and (d) by 1 meter. Additionally, the vertical, red line represents the time of landfall of TY Usagi at Vung tau.

## 4.4 Inland drivers: Rainfall effect on river dynamics

As seen in Sect. 4.2, TY Usagi brought heavy precipitation over the region. In order to better understand the effect of this event on river discharge we map the Saigon system, the topography of the region, the land cover and the spatial distribution of rainfall (Fig. 8a-f). The watershed, urban center and water level stations are also shown in Fig. 8(a). The DEM (Fig. 8b) represents the flatness of the region with the middle, lower and urban parts of the watershed sitting well below 50 m amsl. Further, the lower and urban parts are mostly at an altitude below 10 m amsl (see Fig. 10a). In fact, the slope of the water surface between the upstream and urban water stations is of the order of $10^{-5}$. The distance between these stations is 20 km in a straight line and 35 km along river.

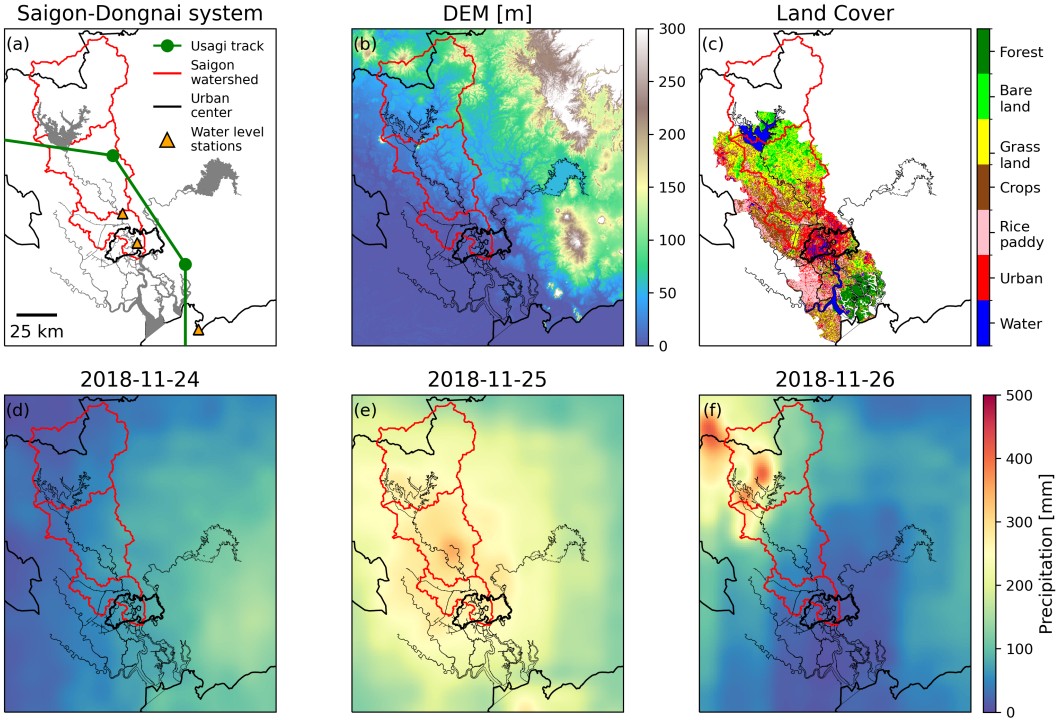

**Figure 8.** (a) The Saigon - Dongnai system, TY Usagi trajectory and the Saigon watershed. The watershed is split into 4 parts: upper, middle, lower and urban; (b) Digital Elevation Model (DEM) of the region. The SRTM vertical datum is global mean sea level and is based on the WGS84 Earth Gravitational Model (EGM 96) geoid (EROS, 2018); (c) Land use map of the region; (d-f) MSWEP daily precipitation over the three days of heavy precipitation connected to TY Usagi.

In terms of land cover (Fig. 8c) , the middle part of the watershed is composed of bare land and grass land in its northern half; of rice paddys, crops and grass land southwest of the Saigon; and of urban cover, crops and rice paddys southeast of the river. However, only about 15 % of this part is urbanized cover (Nguyen et al., 2022). The lower part of the watershed is mainly

covered by crops and grass land west of the river and urbanized land to the east. The urban part is mainly urbanized land cover and waterways with the southeast corner with some grass land (see Fig. 10c).

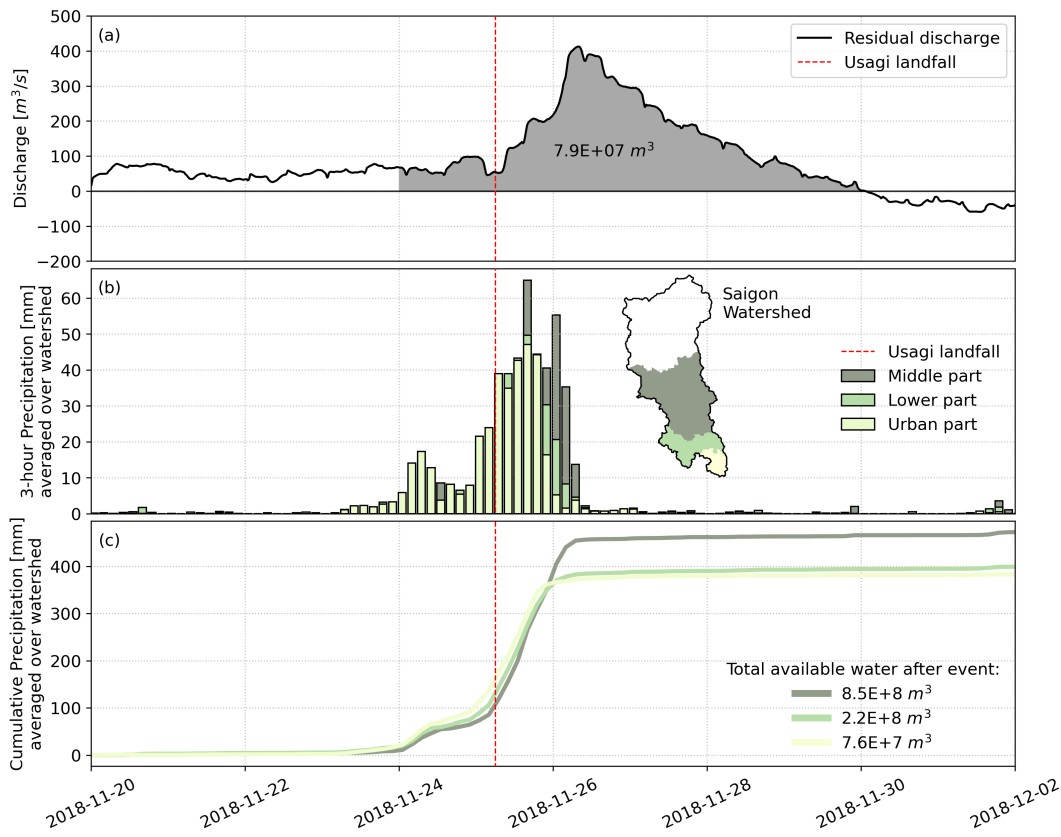

**Figure 9.** (a) Estimated residual discharge at Phu Cuong (upstream) and total volume output during TY Usagi; (b) MSWEP 3-hour precipitation averaged over the middle, lower and urban parts of the watershed. The upper part of the watershed is not considered since all runoff is directed towards the Dau Tieng reservoir. Additionally, the Saigon watershed is delineated in black; (c) MSWEP cumulative precipitation averaged over the middle, lower and urban part of the watershed as well as total available water after the event.

In Fig. 8(d-f), the daily cumulative precipitation is plotted over the Saigon-Dongnai system. On November 24th, 2018 (the day before landfall, Fig. 8d) we see a spatially homogeneous precipitation over the whole watershed of about 100 mm. The following day Usagi made landfall at 06H UTC. Figure 8(e) shows that the bulk of the rainfall happened in the middle of the watershed with some locations with values above 300 mm. The lower and urban parts also witnessed heavy rainfall between 200 and 300 mm. On November 26th, 2018 Usagi was already over Cambodia. The bulk of the rainfall occurred on the upper 480 part of the watershed. The northern part of the middle part of the watershed saw levels of precipitation above 100 mm while the lower and urban parts saw values below 100 mm.

The discharge results during the period of TY Usagi are presented in Fig. 9(a) and the MSWEP precipitation summed over the watershed in Fig. 9(b). Residual discharge starts increasing after landfall whereas the precipitation event starts about 1 day before landfall on Nov. 24th, 2018. Additionally, peak precipitation occurred 10 hours after landfall (16H UTC Nov. 25th, 2018) with a volume of 65 mm over 3 hours and peak discharge occurred about 26 hours after landfall (8H UTC Nov. 26th, 2018) and thus, with a time lag of 16 hours after peak precipitation.

The residual discharge goes to zero about 41 hours after the precipitation event is over. During this time the Saigon river evacuated $7.9E+7$ m$^3$ of water at the Phu Cuong location where discharge is computed. On the other hand, the middle, lower and urban sub-catchments received a total of 472, 399 and 383 mm of cumulative rain (Fig. 9c). This equates to, respectively, $8.5E+8$, $2.2E+8$ and $7.6E+7$ m$^3$ of available water after the event. We can also see (Fig. 8f) that the upper part of the watershed received heavy precipitation on Nov. 26th, 2018. However, from Fig. 9(a) there is no discharge response of the Saigon river which justifies assuming that the upper part of the watershed is not influencing the river's discharge. Even though a river response to rainfall is seen, the residual discharge provoked by the storm is negligible when compared to the total discharge amplitudes due to the tidal forcing (Fig. 6d). Hence, tidal forcing dictates the Saigon's dynamics even during an unprecedented extreme rainfall event. Nonetheless, the behaviour of the residual discharge follows closely the water level's behaviour with a similar time lag in the hydraulic response (see Figures 9a) and 7c).

## 4.5 Urban flooding

The greatest impact of TY Usagi on the population of the HCMC megalopolis came from urban flooding. In this section we discuss the flooding reports at the urban city center in connection with its topography, land use and rainfall extent.

As can be seen from Fig. 10(a), the HCMC is very flat (below 5 m amsl) with the highest part at the center with an altitude of up to 15 m above msl. This higher topography is west of the urban watershed implying a surface runoff gradient from west to east into the Saigon river. The land cover is mainly impervious concrete (Fig. 10c) with some patches of grass land north of the city center.

In Fig. 10(d), the cumulative precipitation during TY Usagi can be found. We see an increasing South-North precipitation gradient which follows the direction of Usagi after it made landfall. The north of the city saw the highest amount of precipitation (up to 400 mm) and the lower and urban parts of the watershed received between 350 and 370 mm. As seen in Sect. 4.4, this amounts to $8.4E+7$ m$^3$ of available water after the 3 days of the event. By design, the proposed model to estimate discharge is not able to estimate discharge at the urban location. Hence, the direct evacuation by the river cannot be discussed. Nonetheless, the Thao Dien station showed a non-negligible increase in water level (see Sect. 4.2) pointing to a surface runoff effect. Rujner and Goedecke (2016) uses the ABIMO model to show that for the urban cover situation in 2016 the total runoff is 38 % of the input by rain water. Thereafter the surface runoff has a share of 22 % and the infiltration 16 %. These values are indicative of the periodic urban flooding problems following heavy rainfall events in HCMC. For TY Usagi, these percentages translate to $6.4E+6$ m$^3$ of surface runoff and $4.6E+6$ m$^3$ of infiltration over the urban part of the watershed. The values for total runoff (surface runoff and infiltration) represent 36.7 % of the total volume discharged ($7.9E+7$ m$^3$) at Phu Cuong after the event. This

provides an indication that, contrarily to the upstream situation, the Saigon river might be capable of evacuating a substantial amount of the rainfall within the few days after the event.

As reported by Vietnamese media, Thao Dien suffered heavy, prolonged flooding with some wards under water for over 24h. The spatial extent and depth of flooding can be found in Fig. 10(b). Indeed, the Thao Dien location shows the highest density of flood points but also the highest depths of flood (up to 0.8 m). Furthermore, flood depths of up to 0.4 m can be found in
the highest elevations of the urban center namely, on the west of the urban watershed which in principle creates runoff in the direction of the Saigon river and thus, towards the Thao Dien location.

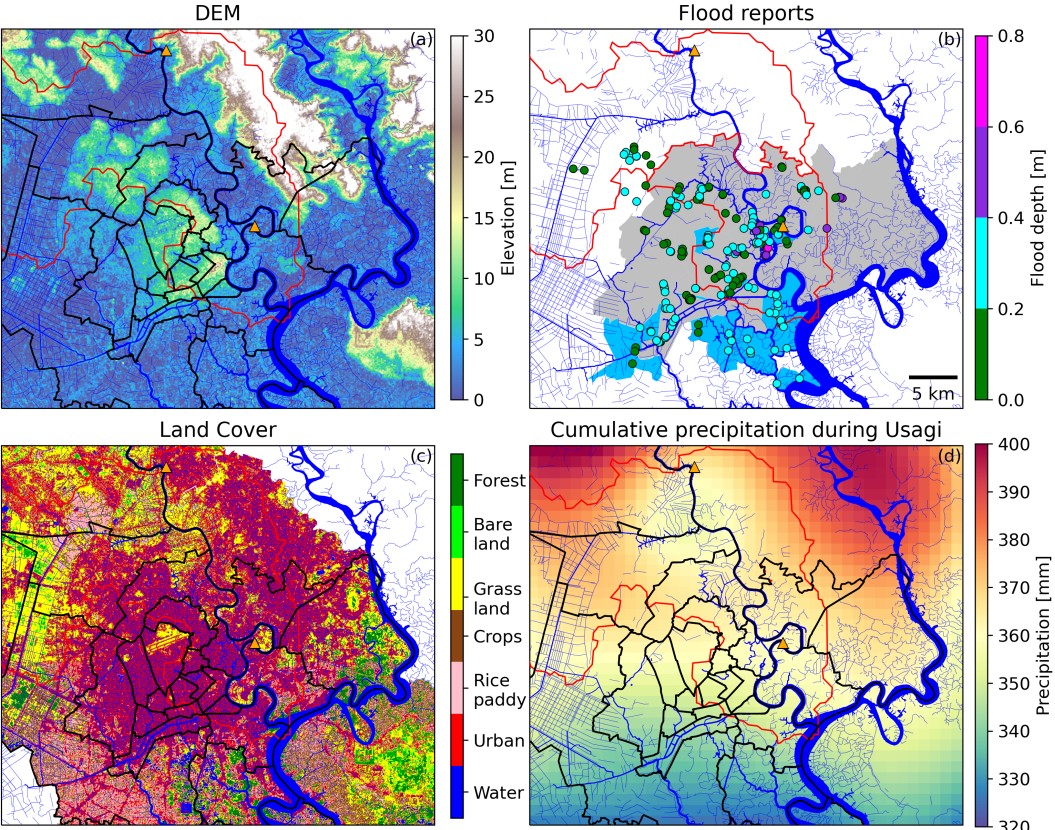

**Figure 10.** (a) Digital Elevation Model (DEM) of the urban city center. The Phu Cuong (upstream) and Thao Dien (urban) water level stations are indicated as orange triangles. The lower and urban parts of the Saigon watershed are delineated in red; (b) Flood extent and depth reported on November 25th, 2018; (c) Land use map of the urban city center; (d) Cumulative MSWEP precipitation during TY Usagi over the three days of heavy precipitation connected to TY Usagi (Nov. 24th-26th, 2018).

The observed time lag in water level increase at the Thao Dien station (Fig. 7c, grey) is compatible with the reported flooding locations and duration. Additionally, both peak rainfall and high tide (Figures 7(b) and 9b) coincided with the reported flooding duration. The high tide and heavy rainfall effectively removed the possibility of surface runoff to flow towards the river and

flooding occurred around Thao Dien. This comes to show that despite the lack of storm surge at the fluvial-marine transition, the coastal tidal forcing is still a main player in the dynamics of urban flooding in HCMC even during TY Usagi.

## 5  Discussion

Coastal regions are often the zones with the highest concentration of population and economic advancement across numerous nations. Ironically, they are also positioned as the most exposed areas to the potential threat of compound floods, arising from
the conjunction of intense precipitation and storm surges. This increased vulnerability can be attributed to the combined factors of dense population, substantial property density, and the inherent risk of storm surges (Shen et al., 2019). Within this study, we present a framework that holds potential for application across coastal urban centers grappling with limited data.

The correlation between precipitation and storm surges is bound by several factors, including meteorological conditions and the topography of the region. TYs often provide the required extreme conditions for the occurrence of compound flooding
events. and are one of the most important triggers of such events in coastal regions (Xu et al., 2022a). Even though compound floods have been receiving attention in recent years, few studies exist on the interplay between compound flood drivers during an historical TY in HCMC (Couasnon et al., 2022; Scheiber et al., 2023). This research contributes to this body of knowledge by analyzing and quantifying the driving factors of the compound flood event due to TY Usagi. On the other hand, it is worth noting that the length of the records of observational data used in our study are relatively short and thus, a statistically-based
study is not possible. However, our fortunate circumstance of having sensors operational before, during, and after the event within this data-scarce region has facilitated the presentation of this case study.

In this study, we found that most precipitation datasets can approximate rainfall patterns over HCMC. Furthermore, when it comes to accurately portraying TY induced rainfall MSWEP showed the best capabilities by being able to represent rainfall quantity and spatial extent which is in line with the literature (Prakash, 2019; Fernández-Alvarez et al., 2020; Tran et al.,
2023). It was found that the worst performing dataset over this domain is ERA5. ERA5 presented large values of RMSE and MBE and low linear correlation which indicate that ERA5 is overestimating rainfall over the whole domain. This finding is in line with current literature: Lavers et al. (2022) found that the largest ERA5 errors are in the Tropics and that ERA5 presents a general wet bias. Additionally, it was found that ERA5 can capture locations and patterns of extreme events but it cannot model the observed precipitation totals. Jiang et al. (2021) found that ERA5 has difficulties in accurately detecting moderate
and high daily precipitation events (above 10 mm/day) over mainland China. It also found that in relatively wet climate such as in the tropical climate zone ERA5 has higher RMSE than satellite-based precipitation products. Indeed, satellite-based products perform generally better than model-based products in low latitudes (Xu et al., 2022b). Rivoire et al. (2021) also found that while ERA5 and CMORPH products would agree over the mid-latitudes, they disagreed over the tropics. In fact, reanalysis products usually struggle to resolve precipitation over the tropics especially tropical cyclones and their surrounding
environment. Slocum et al. (2022) analysis showed biases in the ERA5 environmental diagnostic quantities where the most significant discrepancies are observed in the thermodynamic fields. Notably, there is a cold temperature bias in the boundary

layer, which constrains convective instability. Additionally, ERA5's biases in temperature are evident in the upper troposphere and are accompanied by a notable overestimation of relative humidity.

It was found that the increased rainfall during the wet season appears to have minimal impact on water levels at the seasonal timescale. This phenomenon is attributed to the distinctive pattern of increased rainfall during the wet season coupled with a reduction in river water levels at the two river stations. This observation underscores the dominant influence of coastal water levels on river water levels across the entirety of the Saigon river. On the other hand, a marginal rise in discharge is discernible during the wet season. Several arguments could hold: firstly, we see that river levels are generally lower during the wet season due to the coastal forcing. However, the estimated river discharge, which is a function of the slope, is higher in the wet season despite lower river levels. We propose that this is explained by a decrease in river water level everywhere in the river such that the slope is less influenced than the local river levels by this coastal forcing. Thus, we still capture seasonal differences in discharge given that our discharge estimate is a direct function of river slope. This increased discharge is most likely lead by rainfall-runoff given that the upstream dam management decreases its outflow during the wet season and increases it in dry season to mitigate saline intrusion (Ngoc et al., 2014).

For the time scale of the event (3 days) we find a statistically significant river surge but not a coastal surge. Therefore, there was no impact of coastal water level surge and the river surge seems to be due solely to inland drivers such as rainfall. The Saigon river's water level seems to follow the seasonality of the sea level but at short time scales it is impacted by inland factors. This could be partly explained by the apparent lack of storm surge (see Sect. 4.3). This indicates that TY Usagi had an inland effect on the hydrological system rather than a coastal effect. Additionally, from the wind vector maps of Fig. 2(b-d) it can be seen that offshore winds are created at the Vung Tau region due to the direction and angle of approach of TY Usagi. Hence, throughout the event strong onshore winds never occurred which explains the lack of storm surge at the coast (Fig. 2(a) and Fig. 7c). Examining the skew surge results of Fig. 7(d) further motivates a lack of coastal storm surge but a significant river surge (as seen in Sect. 4.2) after landfall. Furthermore, the event reaches land coinciding with a very low coastal water level during a spring tide (Fig. 7b). This result is in line with the study of (Takagi et al., 2014) which modelled storm surge in Vietnam and found that the potential maximum of the storm surge for the southern sections is rather low (0.7 m) due to the S-shape of the country's shoreline and the cyclonic rotation of TYs in this region. The lack of storm surge during TY Usagi is not unprecedented particularly along the Vietnam coast. Trinh et al. (2020) show that even some of the strongest historical TYs that hit Vietnam can show no record of high storm surges in certain locations along the coast of Vietnam, namely near the cities of Da Nang and Quy Nhon. This indicates that more studies following the methodology of this paper could be valuable if applied to other urban hydrological systems in Vietnam or elsewhere in South East Asia.

The analysis of discharge and rainfall results showed a time lag of 16 hours between peak rainfall and peak residual river discharge. Considering that the middle part of the watershed is the main driver of residual discharge at Phu Cuong, this time lag can be partially explained by the very flat nature of the terrain (Fig. 8b). As shown in Fig. 8(c) , the middle part of the watershed has a significant amount of vegetation which intercepts precipitation and slows the movement of water into river channels. Furthermore, wide spread flooding at the upstream of Phu Cuong played a role in delaying the discharge response. An analysis of flood areas was conducted using MODIS satellite daily and 8-daily surface reflectance products (Fig. A1 in

Appendix A). This analysis proved difficult to interpret within the watershed area due to the presence of clouds throughout the event. However, we found evidence of widespread flooding immediately east of the watershed around the river Vam Co Dong which has similar land cover as the middle part of the watershed (Fig. 8c). Another possible driver of the lag time between rainfall and discharge are aquifer recharging processes. In the middle part of the watershed the groundwater is more influenced by rainfall than by river recharge with shallow aquifers being predominantly recharged by heavy wet season rainfall events (Tu et al., 2022). The existence of a time lag between rainfall and discharge at this location after such heavy rainfall might be due to the recharge of the groundwater table functioning as a buffer. Additionally, monitored groundwater levels are generally above the Saigon river's water level creating an hydraulic gradient from the shallow aquifers towards the river (Khai and Koontanakulvong, 2015; Tu et al., 2022). This possibly indicates a groundwater recharge phenomenon followed by a slow spill towards the river over the few days following the event.

The residual discharge goes to zero about 41 hours after the precipitation event is over. Assuming the middle part of the watershed is the principal driver of the residual discharge at Phu Cuong the river evacuated a 9.3 % of all available water. This low percentage of evacuation is partly explained by the important role that evapotranspiration plays in the climate of the region, which can effectively reduce the peak discharge of the river. Rujner and Goedecke (2016) use the water budget model ABIMO to describe the long-term annual means of the total run-off, surface run-off, evaporation and infiltration around HCMC. The results show that surface runoff takes 7 %, infiltration 6 % and the remaining 87 % is evapotranspiration which is in line with our findings.

Throughout this study, it was assumed that the upper part of the watershed has no influence on the hydrological behaviour of the Saigon river. On Nov. 26th, 2018 (the last day of TY Usagi), the upper part of the watershed received heavy precipitation (Fig. 8f). However, we found no discharge response of the Saigon river (Fig. 9a) which justifies assuming that the upper part of the watershed is not influencing the river's discharge. This part of the watershed is regulated by the Dau Tieng reservoir which is managed by Dau Tieng-Phuoc Hoa Irrigation Exploitation Company. According to its design, the maximum capacity of overflow discharge is of 2800 $m^3s^{-1}$. However, during the dry season the maximum output from the dam is 30 $m^3s^{-1}$ and during the wet season it is around 100 $m^3s^{-1}$ with peaks that can go up to 400 $m^3s^{-1}$ during heavy rainfall events (Dinh and Nguyen, 2019). The reservoir has only one available flood route which is the Saigon river. The flood capacity of the Saigon river section at the foot of the dam is much lower than the maximum overflow capacity causing severe downstream flooding every time the discharge through the spillway exceeds 200 $m^3s^{-1}$. Ever since 2013, the province of HCMC ensures that the reservoir flood discharge capacity is kept below 500 $m^3s^{-1}$. However, when heavy rains are predicted the dam's discharge can be increased to levels close to the peak residual discharge found in Fig. 9. It is thus, possible that part of the residual discharge is a direct cause of the dam's discharge policies which would make the evacuation of rainfall by the river even smaller.

The behaviour of the residual discharge follows closely the river water level's behaviour with a similar time lag in the hydraulic response (see Figures 9a and 7c). This is expected as the driver of the discharge computation is the slope (Eq. 2) which is a function of the river water levels at both stations. The relatively weak hydraulic response could be explained by the fact that both stations have a quasi-simultaneous increase in water level which leads to a less steep slope and thus, weaker discharge. This is due to the regional scale of the Usagi-brought precipitation. Thus, the slope should remain relatively

constant and river discharge might be underestimated. Other possible sources of error in the estimation of discharge can be an over sensitivity to the water level measurement error and difficulty in capturing asymmetrical tide dynamics. Therefore, interpretation of these discharge estimations must be made carefully.

## 6 Conclusions

In this paper, we investigate the most severe rainfall event ever experienced in HCMC, Vietnam. It occurred on November 25th, 2018 when TY Usagi directly hit HCMC. During this event, there was more than 300 mm of rainfall over 24h which led to flooding and considerable material damages. In this work, we put forward an evaluation of the compound flood event and its impact on the region by using a set of tools to characterize and quantify both coastal and inland drivers. For the first time in a data scarce region, all hydrological information was gathered and analyzed during an unprecedented extreme event that affected millions of people in the HCMC megalopolis. We go from an hydrological approach to a data analysis and signal processing approach in order to analyze coastal and river water levels, rainfall and urban flood. This approached not only allowed a thorough investigation of the TY's impacts but also allowed new insights on the hydrological behaviour of this region.

From the evaluation of five research-quality precipitation datasets against in-situ measurements, we find that the MSWEP dataset shows the best performance in estimating rainfall over the region of interest and can capture the precipitation behaviour during TY Usagi (R=0.6; MBE=-4.3 mm and RMSE=7.6 mm). Hence, it is used to analyze TY Usagi's rainfall over the watershed. A statistical overview of the dataset shows that the rainfall brought by Usagi constituted the maximum daily rainfall in the MSWEP data series.

We observe that the impact of increased wet season rainfall on river water levels is dwarfed by the seasonal coastal storm surge caused by the wind pattern of the East Asian summer monsoon, that typically lasts from November to April. This translates to higher water levels during dry season and thus, a coastal control of the Saigon river at the seasonal scale. A mono-disciplinary hydrologist approach may have expected a higher water level during the wet season, which underlines the interest of crossing disciplines at the interface between river and ocean. Additionally, it was found that the highly urbanized land cover

and complex urban canal system of HCMC are modulating the river water level which shows consistently higher residual water levels and skew surge than elsewhere.

The main coastal driver related to a TY event is the associated storm surge. At first, no direct observation of the hydrological impact of TY Usagi is possible on the Saigon river and harmonic tidal analysis was required to filter the tidal fluctuations and observe the effects of such an event. For TY Usagi, the analysis of coastal residual water level and skew surge showed that

the storm surge was not statistically significant when compared with the distribution of values during the wet season. The lack of coastal surge is associated with the timing of the landfall, the direction and the angle of approach of TY Usagi. Landfall occurred during a large amplitude spring tide with a very low tide and the TY never produced strong onshore winds but instead offshore and shore-parallel winds. However, the lack of storm surge is mainly attributed to the wind direction during this event. On the other hand, the Saigon river's urban and upstream water level stations show a significant surge which shows that there

was no impact of coastal surge on river water levels and that river surge was due solely to inland drivers such as rainfall. At short timescales and during the sudden, heavy precipitation event, the river and coastal water levels were decoupled.

After landfall, we find that both river levels and discharge start increasing whereas the precipitation event starts about 1 day before landfall. The peak precipitation and peak discharge occurred 10 and 26 hours after landfall, respectively. The very flat nature of the terrain in addition to the significant amount of vegetation create a time lag of 16 hours between peak discharge and peak precipitation. Furthermore, the river evacuates 9.3% of available rain water pointing to wide spread flooding and aquifer recharging processes in the middle of the watershed. In fact, the extreme rainfall brought about a peak discharge that is above the third quartile for the wet season but that is far from being an outlier. Even though, it was not possible to know the dam's discharge policy during Usagi it is clear that part of the residual discharge could be a direct cause of the dam's discharge. Historically, the dam's discharge policy was of increasing outflow in anticipation of heavy rainfall making the river contribution to evacuating rainfall smaller.

TY Usagi released 7.6E+7 $m^3$ of water over the urban watershed over 3 days. The urban river station showed a non-negligible increase in water level pointing to a surface runoff effect. Our estimation says that urban total runoff is 36.7% of the water volume discharged at Phu Cuong and thus, at the urban location the river is able to evacuate the rain water over the few days after the event. Additionally, we find that the Thao Dien ward shows the highest density of reported flood points but also the highest depths of flood (up to 0.8 m). Furthermore, floods in the west of the urban watershed create runoff in the direction of the Saigon river and thus, towards the Thao Dien water level station explaining its consistently higher surges than in the upstream station. Time lag in river surge and peak precipitation are due to the high tide removing the possibility of surface runoff towards the river. The river water levels were high and rainwater would not effectively drain towards the river but rather linger in the impervious streets of HCMC. In fact, during spring-tides it is common to have river-induced, short-lived flooding in the lower elevation areas of the city, namely in Thao Dien. Adding to this an extreme, persistent precipitation caused widespread flooding. Hence, river flood is delaying the surface run-off and prolonging the flood residence time. This shows that despite the lack of storm surge the astronomical tide forcing is still a main player in the dynamics of urban flooding in HCMC even during TY Usagi.

The methodology presented in this paper encompasses a data processing and analysis work applied to a complex, urban estuarine system. Its foundation lies on the correct choice of TY compound flood drivers and gathering of relevant data in order to characterize the response of the hydrological system to an extreme event. This methodology could easily be applied to any other urbanized estuary both in South East Asia and elsewhere in the world by tailoring the choice of impact factors to the region of interest. Additionally, the extreme event need not be a TY but could be any other event that provokes compound flooding and impacts the respective communities. However, in this study fortunate circumstances allowed us to observe the impact of TY Usagi on the Saigon river system, as our sensors were actively recording during its landfall. River water level measurements at such high time resolution are not generally available in data-scarce regions. Hence, efforts to work together with local researchers and authorities in order to develop data monitoring strategies is crucial for studies of this type in other data-scarce regions. A way to mitigate this problem is by using open-access data such as global precipitation datasets. However, this type of data comes with inherent data quality issues and uncertainties (Scheiber et al., 2023). Another limitation of this

study is the adapted Manning-Strickler equation used to estimate discharge which, even though previously validated for the Saigon river (Camenen et al., 2021), assumes uniform flow for a tidal river. Additionally, if both river water level stations feel an increase in water level the equation will not translate this t-o increased discharge as the surface slope remains constant. Nevertheless, the equation behaves rather well representing the river discharge throughout the monitoring time and during TY Usagi.

Future prospects would be high resolution modelling of the city's flooding regimes during TY events even though models of such complex hydrosystems have expected limitations. Additionally, simulating high numbers of TYs around the area to find TY characteristics that would be more damaging to HCMC could be an interesting follow up. This would allow early warning depending on early developing storm characteristics and a better understanding of expected damages. Thus, providing the city with better information for impact mitigation. Finally, a better understanding of the upstream area of HCMC and the interaction 705 between river discharge, rainfall and groundwater is required as well as a better understanding of the canal network dynamics.

## Appendix A: MODIS satellite study

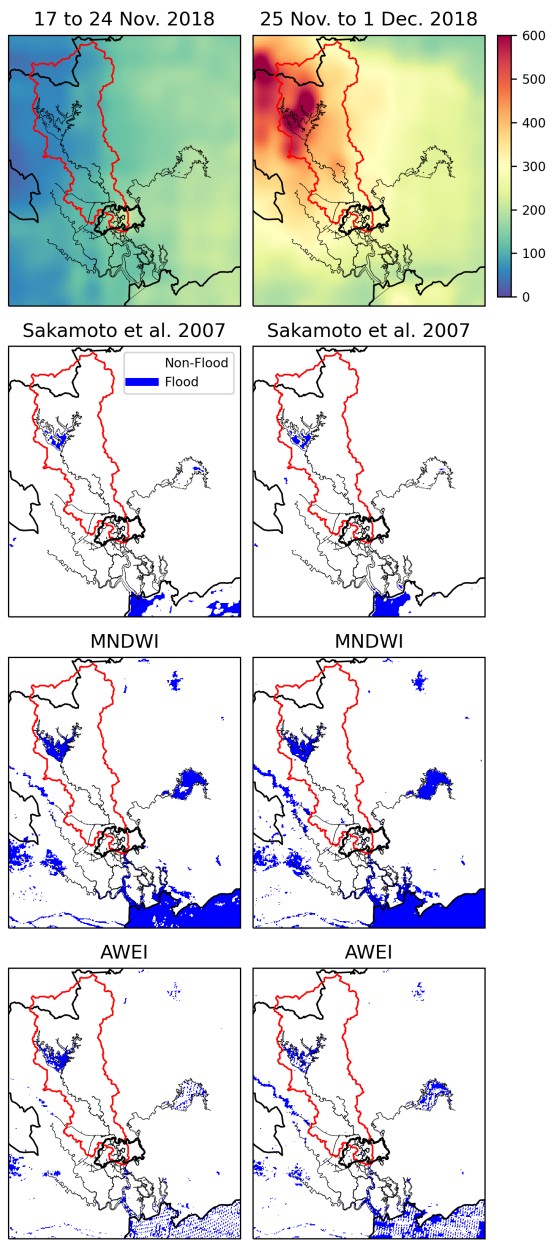

**Figure A1.** DEM, land cover, weekly cumulative precipitation and results of flooded area over the region of interest using MODIS surface reflectance 8-day product. The left column represents the week before the event and the right column the week after. Three methodologies are used to find inundated pixels: Sakamoto et al. (2007); Modified Normalized Difference Water Index (MNDWI); and Automated Water Extraction Index (AWEI).

## Appendix B:  Performance metrics for MSWEP during TY Usagi

**Table B1.** Performance metrics for MSWEP over the month of November 2018 over 6 rainfall gauges in HCMC.

| Rainfall Gauge | RMSE[mm/day] | MBE [mm/day] | R [ ] |
|---|---|---|---|
| Phuoc Long A | 7.2 | 20.2 | 0.62 |
| Binh Chieu | 4.2 | -7.8 | 0.71 |
| Ly Thuong Kiet | 17.3 | -20 | 0.8 |
| An Lac | 11.7 | -15.8 | 0.77 |
| Binh Hung Hoa | 1.3 | -3.5 | 0.11 |
| Cu Chi | 4.2 | 1.2 | 0.64 |
| **Average** | 7.6 | -4.3 | 0.6 |

## Appendix C: Statistical description of datasets

**Table C1.** Statistical description of water level data for the period from January 2017 to December 2018 at the three stations under study. The mean water level over the full period was removed from the series. All non-specified units are in meters.

| Station | Vung Tau | | Thao Dien | | Phu Cuong | |
|---|---|---|---|---|---|---|
| Distance from coast | 0 km | | 60 km | | 80 km | |
| Frequency | 1 h | | 10 min | | 10 min | |
| Season | Dry | Wet | Dry | Wet | Dry | Wet |
| Count of data points | 8224 | 8774 | 48312 | 52993 | 48312 | 52993 |
| Mean | 0.13 | -0.01 | 0.14 | -0.12 | 0.1 | -0.09 |
| Standard deviation | 0.8 | 0.83 | 0.73 | 0.79 | 0.59 | 0.65 |
| Minimum | -2.35 | -2.91 | -2.01 | -2.52 | -1.8 | -2.12 |
| 25th percentile | -0.40 | -0.64 | -0.39 | -0.69 | -0.29 | -0.5 |
| 50th percentile | 0.31 | 0.07 | 0.26 | -0.03 | 0.2 | 0.01 |
| 75th percentile | 0.76 | 0.55 | 0.75 | 0.52 | 0.58 | 0.43 |
| Maximum | 1.61 | 1.47 | 1.48 | 1.4 | 1.15 | 1.11 |

**Table C2.** Statistical description of residual water level for the period from January 2017 to December 2018 at the three stations under study. All non-specified units are in meters.

| Station | Vung Tau | | Thao Dien | | Phu Cuong | |
|---|---|---|---|---|---|---|
| Distance from Coast | 0 | | 60000 | | 80000 | |
| Season | Dry | Wet | Dry | Wet | Dry | Wet |
| Mean | 0.14 | 0.02 | 0.12 | -0.02 | 0.11 | 0.003 |
| Standard deviation | 0.16 | 0.21 | 0.16 | 0.25 | 0.17 | 0.21 |
| Minimum | -0.33 | -0.73 | -0.75 | -2.35 | -0.46 | -0.86 |
| 25th percentile | 0.02 | -0.14 | 0.01 | -0.16 | -0.01 | -0.14 |
| 50th percentile | 0.13 | 0.01 | 0.12 | -0.003 | 0.1 | -0.01 |
| 75th percentile | 0.25 | 0.17 | 0.23 | 0.16 | 0.22 | 0.15 |
| Maximum | 0.76 | 0.76 | 0.91 | 0.94 | 0.93 | 0.93 |

**Table C3.** Statistical description of computed discharge for the period from January 2017 to December 2018 at Phu Cuong.

| | Total Discharge [m^3/s] | | Discharge due to tide [m^3/s] | | Net Discharge [m^3/s] | |
|---|---|---|---|---|---|---|
| Season | Wet | Dry | Wet | Dry | Wet | Dry |
| Count | 52993 | 48305 | 52993 | 48305 | 52993 | 46211 |
| Mean | 242.05 | 135.41 | 60.17 | 66.13 | 181.9 | 68.78 |
| Standard Deviation | 940.96 | 937.86 | 928.96 | 884.64 | 150.09 | 118.03 |
| Minimum | -1626.45 | -1513.89 | -1454.15 | -1421.38 | -352.15 | -535.28 |
| 25th Percentile | -777.9 | -862.25 | -924.57 | -868.44 | 88.57 | 19.37 |
| 50th percentile | 632.81 | 476.49 | 397.32 | 385.62 | 158.95 | 65.94 |
| 75th Percentile | 1112.16 | 1042.00 | 970.93 | 923.61 | 267.18 | 117.24 |
| Maximum | 2482.67 | 1630.29 | 1376.30 | 1302.63 | 1769.07 | 987.94 |

*Author contributions.* Conceptualization, investigation, data collection and curation: Francisco Rodrigues do Amaral; Formal analysis, writing and editing: Francisco Rodrigues do Amaral; Reviewing and supervision: Nicolas Gratiot, Thierry Pellarin.

*Competing interests.* The authors declare having no competing interests.

*Acknowledgements.* This research was conducted thanks to the financial, technical and human support of the CARE-RESCIF initiative (http://carerescif.hcmut.edu.vn/) within the International Joint Laboratory LECZ-CARE. In addition, we would like to thank our colleagues at the DECIDER project for their help in obtaining data. We would also like to thank Dr. Benoit Camenen for his meaningful input on the first draft of this manuscript.

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
