# Peer review of "Assessing Typhoon-induced compound flood drivers: a case study in Ho Chi Minh City, Vietnam"

_EGUsphere, 2023_

## Referee Comment (RC2)

**Summary**

The manuscript evaluates the compound inundation in Ho Chi Minh City in Vietnam during Typhoon Usagi. Their main purpose was to determine which flood mechanism drives the flood along the Saigon-Dong Nai river system. The analysis was performed by analyzing observed data and remote-sensing products. Their finding suggests that the estuary system is mainly dominated by coastal processes, despite the fact that the typhoon event only brought rainfall inundation.

**General Comments**

The manuscript presents a challenging problem to assess in a data-scarce region subject to extreme hazard events, especially the interaction between coastal and hydrologic processes. I have mixed feelings if this manuscript has "enough novelty" to be accepted in a peer-review journal. Since the authors did not develop a new technique or method to investigate the proposed issue, and the results are very specific to this region. Thus, they do not either present broader and general results for the region. However, the compound flood assessment is in high demand, and this manuscript could be a good resource in the literature once it goes under a major revision.

First, there is a lack of novelty in the manuscript, not in the approach selected. In the current version of the manuscript, the novelty of applying the skewness of surge to determine which flood driver dominates it is not highlighted enough, for example. Like this, several other components of the methods are "novel enough" to be published but need to get more attention in the introduction. Thus, I highly recommend including a literature review in the manuscript that summarizes other studies that have used similar techniques to the authors and identify the missing gaps of previous works and how this manuscript tries to fill them.

Second, the authors should focus the theme of the manuscript on a "compound flood assessment" rather than a study of the hydraulic/hydrology response of the watershed. The authors are underselling their work and should put more emphasis on the "hot topic" of compound floods, which is, in reality, what the authors are doing since they are also considering coastal processes and their impacts. I strongly suggest rewriting and refocusing on this theme, including the title.

Third, the manuscript format can be improved substantially to follow a "storytelling" rather than a report. For example, the authors have a "Results and Discussion" section, but a "Discussion" section follows this one. The "Discussion" section is, in reality, a sub-section of the results since they focus on the flood impacts at the urban center, whereas the discussion section should be for comparing their results with previous findings and the physics. The authors did a great job discussing their results in the "Results and Discussion" section. Thus, I strongly recommend separating the discussion from the "Results and Discussion" section and making it a stand-alone section called "Discussion". In addition, the current "Discussion" section should be a sub-section on the new "Results" section.

Lastly, there needs to be a more coherent nomenclature and wording with the current published studies within this field. This could be from a translation from their native language to English. For example, the authors used the word "continental" to refer to hydrologic effects on the flood. However, current studies use the word "inland" more to differentiate from the coastal process in a compound flood event. Thus, the authors assess the "inland and coastal effects" on the hydrosystem, not the "continental and coastal effects". Similarly, the word "evacuate" is being used oddly for the field when referring to the

riverine water leaving its banks and flooding the community. Also, the authors used the term "extreme water levels events", whereas the community uses more "extreme flood events".

**Specific Comments**

- L24: remove the word "coastal" from "coastal engineers" since it can also help water resources engineers. I will also remove the word "reliable forecasting" there is a lot of effort needed to get to this point, such as computational resources, meteorological forecast inputs, accurate models, and not just the basic understanding of the hydrodynamics of the system.
- L25: researchers almost never do decision-making activities, as this statement suggests.
- L35: give an example of population density from another major city (e.g., New York, Hong Kong, Mumbai, etc.) so the reader can have a fair comparison for this statement.
- L55: be consistent with your acronyms. The authors first used LECZ to refer to a low-elevation coastal zone, but in this statement did not use the acronym. Similarly happens with HCMC throughout the entire manuscript.
- L60: quantify the "short spatial scale". Give an example.
- L64: describe what it means to have a negative discharge value on this gauge.
- L68: Where the tides dominate in the river? Until what river length from the outlet or it is complete?
- Figure 1: Need to add a map that shows where HCMC is within Vietnam and then zoom into the basin and the city. Panel (a) add the label for the Vietnam-Cambodia border and the name of the main rivers. What are the grey lines in panel (b)? need to add it to the legend.
- L85-91: the authors give too many details about the classification of the typhoon in this paragraph. I would condense this since it is not pertinent to the manuscript.
- L94: did the authors consider soil type? They only have datasets of topography and land use, but they talk about infiltration and groundwater recharge as one of the main processes during the flood but do not talk anything about the soil types which govern these processes.
- L95: as the statement is written, it says that extreme events, like a typhoon, would have an effect on the astronomical tides. However, they do not alter this response.
- Figure 3: all the components in the diagram are talked about in the main text of the manuscript, with the exception of the "mapping and characterization". The authors should explain this more. Also, on the figure label, the focus is on the "hydrological system", but it also talks about coastal processes. I recommend changing the wording toward "estuarine system" which implies both coastal and hydrologic processes.
- L110: the authors lack a justification for the selection of a 3-day rainfall total for this analysis. All the datasets have a maximum daily time scale. Why not select a daily accumulation rather than a 3-days total? Also, the word "adequate" needs a quantification. What is adequate for the authors might not be for other readers.
- L112: What criteria the authors used to "deem sufficient" the quality of the observed data?
- Table 1: the nomenclature for the correlation coefficient equation is missing. What represents "cov(P,O)"?
- Table 3 is in the text before being cited. The table should be cited first and then shown. Also, how can the authors visualize a semi-diurnal tidal behavior if the time resolution of the tidal gauge has a daily time step, meaning only one value per day?

- L173: what was the time window for the moving average performed for the monthly tide values?
- L177: mention the amount of tidal constituent used in the resynthesize analysis.
- L185: generally, you should not refer to a figure before presenting other ones. For example, the authors cite Figure 7, but only have presented three figures.
- L216-217: the authors should justify why they used the selected thresholds of dH and dt.
- L260-261: have other studies found similar results with ERA5?
- L303-304; L320-322: are these findings also been found by other researchers? Find additional literature that supports or refutes your findings. That should be part of your new discussion section.
- Figure 6: why the observed discharge is higher in the wet season than in the dry if the observed water level is higher in the dry season than in the wet? Discharge is computed from the water level, so they should have the same behavior, which is not the case.
- L343: where the coastal water level is the main driver and not the rainfall?
- L381-387: move out from results into methods and data collection. This will explain to the reader why the authors also consider wind data. It was quite strange when I saw wind vectors in Figure 2.
- Figure 7: add a legend to the figure explaining each color of the lines. Also, add the datum to which the levels are referenced.
- L410-424: these are not results and more a description of the study area. I would move them out and into the study area section, including the figures. Maybe the wind vector panels in figure 2 can be swapped with the top three panels in Figure 8.
- Figure 8: panel a) the track line in the legend is green but in the map is purple. Panel b) add the datum of the elevation from the DEM.
- Conclusion: Add a paragraph about the limitation/assumption the method used by the authors may have.

---

## Author Comment (AC1)

We highly appreciate and are very thankful for the time and effort that was invested in reviewing our manuscript. Thank you for initiating this discussion. After carefully studying the constructive queries and comments, we have thoroughly revised our manuscript in an attempt to clarify as much as possible the content of the draft manuscript. Below you will find our comments (in blue) to your feedback (in **black**, **bold**).

**General comments:**

**In this paper the authors describe the impact of Typhoon Usagi rainfall on the hydrology around Ho Chi Minh City, Vietnam on 25th November 2018. In particular they evaluate the impact of intense rainfall and storm surge, on the hydrological system using tools to characterize these coastal vs surface runoff ('continental') contributions. The authors contrast the river, surface runoff and coastal (tidal) responses at two river gauge stations upstream of the Typhoon-struck coastline. The preprint is well referenced and structured, with good quality figures and tables to support the key messages. The hypothesis is given, but objectives have not been stated clearly. References are occasionally missing to back up statements in the text. Some technical terms and conclusions should probably be explained further for a general audience to avoid confusion. Some of the technical aspects in the method (and conclusions drawn from this) were difficult to follow so perhaps would benefit from being clarified/rewritten (e.g. surface runoff assumptions downstream of a dammed area). The standard of English is good, the title reflects the contents of the manuscript also. While the abstract could more concisely describe method and results, in the main body of the paper the methodology, the presentation of results, and the conclusions reached are all satisfactory. I have not checked the statistics contained in the appendix tables - these are accepted 'as is'. The manuscript requires some editorial assistance from NHESS (some grammar errors noted).**

**However, the approach taken in this paper is an interesting one, and therefore I believe the manuscript will ultimately contribute something new to the scientific discourse. I recommend accept subject to (quite a few) minor revisions as described below:**

Thank you very much for your contribution. The aspects to be improved found in your comment will be considered in the revised manuscript. The objectives of the paper will be stated more clearly in addition to the hypothesis. We will review and strengthen statements that were found to be lacking reference. Overall, we will take your comments below and implement them in order to make the draft as clear as possible in the revised manuscript.

**Minor revisions:**

**While the Aim of this manuscript is clear (to investigate the precipitation and storm surge impacts from Typhoon Usagi on the local hydrology of the Saigon river, HCMC), the manuscript would benefit from having the objectives clearly stated in the introduction section too (L46-51).**

Thank you for this comment. The objectives of this manuscript are as follows:

1. To provide a multi-approach methodology based on distinct sources of data to characterise the hydrologic response of a tropical, urban tidal river to a typhoon.
2. To better understand which of the potential contributors to urban flooding (precipitation, storm surge or river flood) were most relevant during this particular event.

3. To create a holistic picture of how the different parts of the hydrological system (terrain elevation, land cover, precipitation, tidal river and coastal surge) contribute to the response of the river to this extreme event as well as to the flooding experienced by HCMC residents.

We will clearly define the objectives of the manuscript in the revised manuscript.

**It would benefit the paper to be clearer with terminology, from the beginning of the manuscript, and to use it consistently throughout. Some examples:**

**1. A cleaner differentiation between river levels and sea levels (and river gauge vs tide/sea gauge stations). The phrase 'water levels' is a little generic even when discussing data from around a tidally influenced river /estuary.**

Indeed. We will standardize the terminology throughout the manuscript. Thank you.

**2. What is H / water level? It is not stage (with a datum from the river gauging station), seemingly. Is it depth of water above the (unknown) channel bed level, or head?**

Thank you for your comment. We utilize the term 'water level ($H$)' to refer to the height of the column of water above the pressure gauges in the river. The tidal oscillations are propagated from the coastal tide gauge to the river gauges. This is the signal that is captured by the gauges and that we call 'water level ($H$)'. Since there is no fixed datum between river and tide gauges, we normalize all signals by mean removal. This makes the tidal harmonics to oscillate about zero for all gauge locations thus, making them comparable. In addition, the $dH$ parameter in the equation to estimate discharge (Eq. 2 in the manuscript) provides an additional calibration parameter that helps mitigate this problem. We will make this clearer in the revised manuscript.

**3. In section 3.6: Water discharge is a phrase that doesn't translate well - do the authors mean river (fluvial) discharge?**

Yes, we mean river (fluvial) discharge. We will use this terminology in the revised manuscript. Thank you.

**It would benefit the paper to support particular statements with more references. E.g.,:**

**- L 30 "Vietnam lies within the most active cyclogenesis regions in the world".**

Thank you for this remark. The western North Pacific which includes the South China Sea is the most active basin of cyclone activity in the world. We will further support this statement using a selection of the following references:

1. Gao, S., Zhu, L., Zhang, W. et al. Western North Pacific Tropical Cyclone Activity in 2018: A Season of Extremes (2020). Science Reports 10, 5610 https://doi.org/10.1038/s41598-020-62632-5
2. Klotzbach, P. J., Wood, K. M., Schreck, C. J., Bowen, S. G., Patricola, C. M., & Bell, M. M. (2022). Trends in global tropical cyclone activity: 1990–2021. Geophysical Research Letters, 49, e2021GL095774. https://doi.org/10.1029/2021GL095774
3.

Feng, X., Klingaman, N.P. & Hodges, K.I. Poleward migration of western North Pacific tropical cyclones related to changes in cyclone seasonalit (2021). Nat Commun 12, 6210. https://doi.org/10.1038/s41467-021-26369-7

4.

Ruifen Zhan, Ming Ying, Peiyan Chen, On Tropical Cyclone Activity Over the Western North Pacific in 2012 (2013), Tropical Cyclone Research and Review, https://doi.org/10.6057/2013TCRR01.04.

**- L38 & L74-76. HCMC is one of the most vulnerable coastal regions in the world to flooding: Why does it rank most vulnerable (More of a certain type of flood hazard than other LECZs? A greater population at risk? More likely to /higher frequency of flooding than other locations?)? It has already been stated that the probability of typhoon occurring in southern Vietnam is not large (L33).**

Thank you for your comment. Ho Chi Minh city is often presented as one of the most vulnerable cities in the world with respect to climate change. Some of these vulnerabilities are water-related issues such as lack of urban services like drinking-water management,  sanitation and rainwater drainage, In particular, flooding vulnerability is linked to sea level rise, rainfall intensification and ground subsidence, which can reach 0.02 m/year (in some geological areas),  while 65% of the city is located at less than 1.5 m above sea level. In addition, HCMC is home to almost 10 million inhabitants and its population grows at about 3% per year with these risks posing a threat to many livelihoods. The urban growth rate (about 16 km$^2$ per year since 2000) is also an important factor as imperviousness of soils reduces infiltration potential and increases the flood risk (UNESCO Water, megacities and global change: portraits of 15 emblematic cities of the world, 2016).

Several studies that consider HCMC as a hotspot of vulnerability to climate change can be found in literature such as

1. Nicholls, R. J. (1995). Coastal megacities and climate change. GeoJournal, 37(3), 369-379.
2. Dasgupta, S., Laplante, B., Meisner, C., Wheeler, D., & Yan, J. (2007). The impact of sea level rise on developing countries: a comparative analysis. Climatic Change, 93(3-4), 379-388.
3. Nicholls, R. J., Wong, P. P., Burkett, V. R., Codignotto, J. O., Hay, J. E., McLean, R. F., … & Woodroffe, C. D. (2007). Coastal systems and low-lying areas. Climate Change 2007: impacts, adaptation and vulnerability. Contribution of Working Group II to the Fourth Assessment Report of the Intergovernmental Panel on Climate Change.
4. Carew-Reid, J. (2008). Rapid assessment of the extent and impact of sea level rise in Viet Nam.
5. Webster, D., & McElwee, P. (2009). Urbanization dynamics and policy frameworks in developing East Asia.
6. ADB (2010). The economics of climate change in Southeast Asia: a regional review.
7. Birkmann, J., Garschagen, M., Kraas, F., & Quang, N. (2010). Adaptive urban governance: new challenges for the second generation of urban adaptation strategies to climate change. Sustainability Science, 5(2), 185-206.
8. Fuchs, R.J.(2010) Cities at Risk: Asia's Coastal Cities in an Age of Climate Change
9. Fuchs et al.(2011) Floods in Megacities: A case study of vulnerabilities and response capacities in Metro Manila 10.Hanson et al.(2011) A global ranking of port cities with high exposure to climate extremes

We will clarify why HCMC is especially at risk in the revised manuscript and provide the relevant sources.

**- L47-48 Perhaps introduce the concept of/your meaning of the terminology "coastal and continental effects".**

Indeed, a formal explanation of our understanding of "coastal" versus "continental" effects is missing. All typhoon-related phenomena that influences coastal dynamics (such as storm surge, wind, tide) is referred to as "coastal" effects. On the other hand, typhoon-related phenomena influencing the continental hydrology (precipitation, surface run-off, river discharge, flooding) is referred as "continental' effects. Thank you for your comment, we will provide this explanation in the text and connect it with Fig. 3 of the manuscript for clarity.

**- L48-50 the sentence beginning "For the first time in a data scarce region, satellite and in-depth measurements were gathered and jointly analyzed during an unprecedented extreme event ..." might require some qualification for two reasons. Firstly there are gauges and data as shown in Fig 1 (is data scarce because it is incomplete?). Secondly, more generally, there are a number of papers that have combined satellite data with (limited) data collected on the ground in areas which are considered 'data-sparse' and this is often explored through the lens of extreme flood events as case studies. E.g. Dung et al., 2011 (https://hess.copernicus.org/articles/15/1339/2011/), Kuenzer et al., 2013 (https://www.mdpi.com/2072-4292/5/2/687), Mohammed et al., 2018 (https://www.mdpi.com/2072-4292/10/6/885), Tegos et al., 2022 (https://www.mdpi.com/2306-5338/9/5/93 ). Perhaps it is just sentence construction -i.e., it's the first time this new method has been applied, in a "data scarce" region?**

Thank you for this comment. You are correct in that it is a sentence construction issue and the literature provided is very relevant. The message we would like to put across is that the methodology has never been used in such a region (to the best of our knowledge) but also, that the use of multi-source data has never been used in this specific basin namely, the Saigon river basin. We use the term "data-scarce" to refer to the fact that it is very difficult to obtain reliable, free data in this region given that it exists. We will re-phrase this sentence accordingly.

**- L72-74. The Trinh et al., 2020 reference I believe refers to the wider Northwest Pacific Ocean being one of the most active regions of the world for Tropical Cyclones [TCs] (~30% of all annual tropical storms), not the South China Sea region. Many of the NWP TCs don't travel into this smaller area. It would be beneficial to clarify/correct this statement.**

Thank you for this comment. Indeed it is the Northwest Pacific Ocean which includes the South China Sea (also known as East Sea of Vietnam) that is referred as the most active region in the world. We will correct this statement in the text of the revised manuscript.

**– L 74. Please define a typhoon (e.g., wind speeds or category scale) vs a tropical storm.**

Thank you. We use the terms tropical storm, severe tropical storm and typhoon according to the intensity classification of the Japan Meteorological Agency who officially monitors tropical cyclones that occur within the Northern Hemisphere between the anti-meridian and 100°E. The definitions are based on 10-min average maximum wind speed as follows:

| | |
|---|---|
| Tropical Depression | Maximum wind speed < 17m/s (34kt) |
| Tropical Storm | 17m/s (34kt) ≤ Maximum wind speed < 25m/s (48kt) |
| Severe Tropical Storm | 25m/s (48kt) ≤ Maximum wind speed < 33m/s (64kt) |
| Typhoon | 33m/s (64kt) ≤ Maximum wind speed < 44m/s (85kt) |
| Very Strong Typhoon | 44m/s (85kt) ≤ Maximum wind speed < 54m/s (105kt) |
| Violent Typhoon | 54m/s (105kt) ≤ Maximum wind speed |

We will add a source to this information and mention the relevant definitions in the text for the revised manuscript.

**– L127. Technically there are four categories in Table 2, not three.**

Thank you. We will correct this in the new version.

**– L168. Please define 'low net discharge' – i.e. low is relative to what/under what categorization?**

Thank you for this comment. 'Low net discharge' is in comparison with the instantaneous discharge due to tidal fluctuations which are one order of magnitude higher than the net discharge. We will precise this in the revised manuscript.

**– L216. dH is introduced to correct for an unknown datum. How was it derived/calculated?**

This parameter was derived by using a non-linear least-squares curve fitting of equation 1 to two 24 hour ADCP discharge measurements (as in Camenen et al. 2021). In short, we use this technique to find the best fitting possible between equation 1 and discharge measurements while taking into account measurement uncertainty. This effectively minimized the Root Mean Square Error (RMSE) between ADCP measurements and estimated discharge. We will clarify this in the revised manuscript.

1. Camenen, B., Gratiot, N., Cohard, J. A., Gard, F., Tran, V. Q., Nguyen, A. T., Dramais, G., van Emmerik, T., & Némery, J. (2021). Monitoring discharge in a tidal river using water level observations: Application to the Saigon River, Vietnam. Science of the Total Environment, 761, [143195]. https://doi.org/10.1016/j.scitotenv.2020.143195

**- datums generally are unstated throughout this paper?**

Thank you for this comment. As briefly mentioned before, there are no datums that can be used as reference for the water level measurements in the river. The tide gauge is the only one to have a station datum as provided in the repository of the Sea Level Center of the University of Hawaii (link to Vung Tau station datum information: https://uhslc.soest.hawaii.edu/stations/?stn=383#datums . However, given the unknown datum of the river stations we cannot compare them. In order to mitigate this problem with perform mean normalization across the gauges such that the tidal signal is fluctuating about zero, as previously mentioned.

**What is the mean sea level reference datum - Is that local mean or global mean? Also, in L237-240 – the datum could be provided for the SRTM DEM; this is relevant if (river and coastal) flood levels are measured against these elevations.**

Indeed, this information is not present in the text. The SRTM vertical datum is global mean sea level and is based on the WGS84 Earth Gravitational Model (EGM 96) geoid as specified in:

1. U.S. Geological Survey, Earth Resources Observation and Science (EROS) Center. (2018). USGS EROS Archive - Digital Elevation - Shuttle Radar Topography Mission (SRTM) 1 Arc-Second Global. Retrieved from https://www.usgs.gov/centers/eros/science/usgs-eros-archive-digital-elevation-shuttle-radar-topography-mission-srtm-1

Throughout the manuscript when using the term "mean sea level" we refer to the global mean sea level used as datum for the SRTM data. We will add this information in the revised manuscript.

**- L221-224 – the introduction of K, dt and dz parameters is difficult to understand without context, perhaps rephrase this paragraph to clarify why they are important, what they mean and how they are used to optimise RMSE if this is important to your manuscript.**

Thank you for this comment. We will introduce and precise the importance of these parameters for the discharge estimation in the revised manuscript.

- K, the Manning-Strickler coefficient of the river reach is a measure of channel roughness or friction and is assumed constant.
- dt, is a time lag required to account for the propagation of the tidal wave between the downstream location to the upstream location.
- dz (which is a mistake and will be modified in the revised manuscript to read '*dH*'), is used to compensate for the fact that the reference points of each location are different and unknown.

These parameters are the calibration parameters that allow adjustments in the output of Equation 1 such that RMSE can be minimized. We will clarify this in the revised manuscript.

**- Unclear about the statement that river slope explains seasonal variation in discharge rates (L344-346) and lack of discharge response after intense rainfall (L475-479). Perhaps explain a bit more the thinking in these sentences. [Hup-Hdn] should be relatively constant if levels at both locations change by approximately the same amount?**

Thank you for this comment. In both L344-346 and L475-479, the argument is similar.

For L344-346: We see that river water levels are generally lower during the wet season due to the coastal forcing. However, the estimated river discharge, which is a function of the slope, is higher in the wet season despite lower water levels. We propose that this is explained by an overall decrease in water level along the river such that the slope is less influenced than the water levels by this coastal forcing.

For L475-479: Similarly, here we find a quasi-simultaneous increase in river water levels at both upstream and downstream locations. This is due to the regional scale of the Usagi-brought precipitation. Hence, the slope should remain relatively constant (as mentioned in your comment). This is one of the drawbacks of the method used to estimate discharge.

We will clarify the thinking behind these arguments in the revised manuscript.

**L559-560- "…high tide removing possibility of surface runoff to the river". I don't understand this reasoning/sentence. Please explain? Do you mean obscuring the response?**

Thank you for this comment. From figure 7c) and figure 9b), we can see that the peak precipitation coincides with peak river discharge. Additionally, both of these coincide with an asymmetric tide period where high tide is followed by a high-water low tide (see figure 7b). What we mean in this sentence by "removing possibility of surface runoff" is that the river water levels were high and rainwater would not effectively drain towards the river but rather linger in the impervious streets of HCMC. In fact, during spring-tides it is common to have river-induced, short-lived flooding in the lower elevation areas of the city, namely in Thao Dien. Adding to this an extreme, persistent precipitation caused wide-spread flooding. So, we propose that river high water is delaying the

surface run-off and prolonging the flood residence time. We will clarify this sentence in the revised manuscript.

**Fig A1 – colorbar units have been cut off.**

Noted. This will be corrected in the revised manuscript.

Thank you for your comments and corrections, which were very helpful to ensure the correctness of this paper. The invested efforts are much appreciated.

---

## Author Comment (AC2)

RESPONSE TO REVIEW #2

We highly appreciate the time and effort that was invested in reviewing our manuscript. After carefully studying the constructive queries and comments, we have thoroughly answered them in this document in an attempt to clarify as much as possible the content of the draft manuscript and/or the changes that will be made in the revised version where needed. Below you will find our comments (in blue) to your feedback (in **black**) to be implemented in the revision of the draft manuscript.

**Summary**

The manuscript evaluates the compound inundation in Ho Chi Minh City in Vietnam during Typhoon Usagi. Their main purpose was to determine which flood mechanism drives the flood along the Saigon-Dong Nai river system. The analysis was performed by analyzing observed data and remote-sensing products. Their finding suggests that the estuary system is mainly dominated by coastal processes, despite the fact that the typhoon event only brought rainfall inundation.

**General Comments**

The manuscript presents a challenging problem to assess in a data-scarce region subject to extreme hazard events, especially the interaction between coastal and hydrologic processes. I have mixed feelings if this manuscript has "enough novelty" to be accepted in a peer-review journal. Since the authors did not develop a new technique or method to investigate the proposed issue, and the results are very specific to this region. Thus, they do not either present broader and general results for the region. However, the compound flood assessment is in high demand, and this manuscript could be a good resource in the literature once it goes under a major revision.

First, there is a lack of novelty in the manuscript, not in the approach selected. In the current version of the manuscript, the novelty of applying the skewness of surge to determine which flood driver dominates it is not highlighted enough, for example. Like this, several other components of the methods are "novel enough" to be published but need to get more attention in the introduction. Thus, I highly recommend including a literature review in the manuscript that summarizes other studies that have used similar techniques to the authors and identify the missing gaps of previous works and how this manuscript tries to fill them.

Second, the authors should focus the theme of the manuscript on a "compound flood assessment" rather than a study of the hydraulic/hydrology response of the watershed. The authors are underselling their work and should put more emphasis on the "hot topic" of compound floods, which is, in reality, what the authors are doing since they are also considering coastal processes and their impacts. I strongly suggest rewriting and refocusing on this theme, including the title.

Third, the manuscript format can be improved substantially to follow a "storytelling" rather than a report. For example, the authors have a "Results and Discussion" section, but a "Discussion" section follows this one. The "Discussion" section is, in reality, a sub-section of the results since they focus on the flood impacts at the urban center, whereas the discussion section should be for comparing their results with previous findings and the physics. The authors did a great job discussing their results in the "Results and Discussion" section. Thus, I strongly recommend separating the discussion from the "Results and Discussion" section and making it a stand-alone section called "Discussion". In addition, the current "Discussion" section should be a sub-section on the new "Results" section.

Lastly, there needs to be a more coherent nomenclature and wording with the current published studies within this field. This could be from a translation from their native language to English. For

example, the authors used the word "continental" to refer to hydrologic effects on the flood. However, current studies use the word "inland" more to differentiate from the coastal process in a compound flood event. Thus, the authors assess the "inland and coastal effects" on the hydrosystem, not the "continental and coastal effects". Similarly, the word "evacuate" is being used oddly for the field when referring to the riverine water leaving its banks and flooding the community. Also, the authors used the term "extreme water levels events", whereas the community uses more "extreme flood events".

Our initial motivation for writing this manuscript was in light of the previous research conducted by Camenen et al. in 2021 (introduced in L41-L45). In this research, which provided a monthly evaluation of the Saigon River's response to this extreme rainfall, there emerges a paradoxical result of a lack of direct response in both water level and discharge. However, the authors of the current manuscript recognize the importance of investigating the finer-scale behaviour of the hydrosystem during and immediately after this event, as the monthly average may not capture the dynamics adequately. This is the reason why we chose our title to be more generally focused on the hydrosystem. However, as correctly pointed out by Reviewer 2, the manuscript addresses a challenging problem of assessing compound inundation in a data-scarce region. We do believe that the most important outcome of this manuscript becomes exactly this one. We appreciate the suggestion to focus the theme of the manuscript on a "compound flood assessment" rather than solely on the hydraulic/hydrologic response of the watershed. You are correct that our study addresses the issue of compound floods by considering both coastal and hydrologic processes. We will reframe the manuscript to put more emphasis on the compound flood assessment, as this is a hot topic in the field. Additionally, we will change the title for the final revision to be: *"Understanding Compound Flooding in Ho Chi Minh City: Assessing the Combined Impacts of Rainfall and Coastal Processes during Typhoon Usagi".*

We understand the reviewer's concerns regarding the novelty of the study. While we did not develop a new technique or method, we believe our research contributes to the field by applying and adapting existing approaches to the context of this case study. We would like to clarify why we strongly believe the results provided in this manuscript are valuable for the scientific community. The two main points are as follows:

Firstly, for the first time (to the best of our knowledge), we compare and evaluate several distinct datasets of precipitation against in-situ measurements over this region; and, in particular, their capacity of capturing extreme rainfall as brought by Typhoon Usagi. Indeed, the techniques used to obtain this result are not new but we do believe that this result is a valuable contribution to the scientific community interested in this area in particular, but also other areas with similar characteristics. Given the context of data-scarcity this result is especially relevant to the current body of knowledge.

Secondly, we jointly analyse different bodies of free, open-access data (precipitation, water level, topography, land use, wind and flooding hotspots) to holistically characterize the drivers of compound flooding during this extreme event. Each method to analyse the data and provide results is not novel but have never been used in unison for a case study of compound flooding during an extreme event. We believe that the value of this manuscript lies, on one hand, in the smart integration of different sources of data with different analysis methods and, on the other hand, in the outcomes and insights of this integration. We effectively show that even in a region where reliable data is hard to obtain, it is possible to unravel (to a certain extent) the complex interplay between coastal and hydrologic processes and gather meaningful information that allows the establishing of a case study such as this one.

We acknowledge that the results are specific to this region and may not offer broader and general conclusions for other areas. However, the focus on compound flood assessment is highly relevant

and significant, as it addresses the growing demand for understanding the combined impacts of different flood drivers in the region. By analyzing the coastal and hydrologic processes, our study offers valuable insights into the mechanisms driving floods in the HCMC urban, low elevation coastal zone, which is a hotspot of vulnerability. HCMC is often presented as one of the most vulnerable cities in the world with respect to climate change and water-related issues (please see the answer to Reviewer #1 for a more in-depth explanation of the reasons why).

Thank you for pointing out the need for improved manuscript formatting and a clearer storytelling approach. We agree that the current structure could be enhanced to provide a more coherent flow of information. We will make the necessary changes by separating the current "Discussion" section from the "Results and Discussion" section and creating a standalone "Discussion" section. The current "Discussion" section will be incorporated as a subsection in the new "Results" section. This revision will ensure a more logical organization of the manuscript and allow for better comparison of our results with previous findings and underlying physical processes.

We appreciate the comment regarding the nomenclature and wording used in the manuscript. We apologize for any confusion caused by the terminology inconsistency. We will revise the manuscript to use the more commonly accepted term "inland" instead of "continental" to refer to hydrologic effects on the flood, aligning with current studies in the field. Similarly, we will replace the term "evacuate" with a more appropriate term to describe the riverine water leaving its banks and flooding the community. Additionally, we will modify the phrase "extreme water levels events" to "extreme flood events" to align with the standard terminology used in the community.

Finally, we would like to express our gratitude to the reviewer's suggestion that the manuscript could serve as a useful resource in the literature. We will take their feedback into account and ensure that the manuscript undergoes thorough revision, including better highlighting the novelty of our approach, providing a more comprehensive literature review, and emphasizing the contribution of our study to the field of compound flood assessment. Through these revisions, we aim to strengthen the manuscript's value and relevance for publication in NHESS.

**Specific Comments**

• **L24: remove the word "coastal" from "coastal engineers" since it can also help water resources engineers. I will also remove the word "reliable forecasting" there is a lot of effort needed to get to this point, such as computational resources, meteorological forecast inputs, accurate models, and not just the basic understanding of the hydrodynamics of the system.**

Thank you for your remark. We will implement this in the revision.

• **L25: researchers almost never do decision-making activities, as this statement suggests.**

Thank you for your remark. Indeed, we will implement this in the revision.

• **L35: give an example of population density from another major city (e.g., New York, Hong Kong, Mumbai, etc.) so the reader can have a fair comparison for this statement.**

Thank you for your remark. To provide a fair comparison, an example of population density from another major city will be included to enhance the reader's understanding.

• **L55: be consistent with your acronyms. The authors first used LECZ to refer to a low-elevation coastal zone, but in this statement did not use the acronym. Similarly happens with HCMC throughout the entire manuscript.**

Thank you for your remark. Consistency in acronyms will be ensured throughout the manuscript. The authors will consistently use "LECZ" to refer to the low-elevation coastal zone and maintain consistency with the acronym "HCMC" for Ho Chi Minh City.

• **L60: quantify the "short spatial scale". Give an example.**

Thank you for your remark. Here we refer to short spatial scale to mean the scale of the district size of HCMC namely in the order of magnitude of the kilometer. For example, it can be that it is raining heavily in one district whereas in another one it is not raining at all. We will clarify this in the revision.

• **L64: describe what it means to have a negative discharge value on this gauge.**

Thank you for your remark. Having a negative discharge at this location means we have river flow towards upstream. This will be clarified in the next version.

• **L68: Where the tides dominate in the river? Until what river length from the outlet or it is complete?**

Thank you for your questions. Indeed, the tide dominates the totality of the river Saigon from the confluence with the Dongnai river to the outlet of the Dau Tieng reservoir. This information will be included in the revision.

• **Figure 1: Need to add a map that shows where HCMC is within Vietnam and then zoom into the basin and the city. Panel (a) add the label for the Vietnam-Cambodia border and the name of the main rivers. What are the grey lines in panel (b)? need to add it to the legend.**

Thank you for your remarks. A map depicting the location of HCMC within Vietnam will be added. In panel (a), the label for the Vietnam-Cambodia border will be included in the next version and the name of the main rivers as well. The grey lines are depicting the complex natural and artificial canal network around the main rivers. The legend will be updated to explain this.

• **L85-91: the authors give too many details about the classification of the typhoon in this paragraph. I would condense this since it is not pertinent to the manuscript.**

Thank you for your remark. We will take this into account in the revision.

• **L94: did the authors consider soil type? They only have datasets of topography and land use, but they talk about infiltration and groundwater recharge as one of the main processes during the flood but do not talk anything about the soil types which govern these processes.**

Thank you for your remark. Indeed, while datasets of topography and land use are available, the discussion will be expanded to include the influence of soil types on infiltration and groundwater recharge during the flood. We did not fully consider the soil type upstream of the city center in the text but did consider literature concerning this topic (Khai et al. 2015, Tu et al. 2022). Nonetheless, a more thorough literature review on soil type will be added in the revision.

*Khai, H. Q. and Koontanakulvong, S.: Impact of Climate Change on groundwater recharge in Ho Chi Minh City Area, Vietnam, In proceedings: THA 2015 International Conference on Climate Change and Water & Environment Management in Monsoon Asia, https://www.researchgate.net/publication/275643904_Impact_of_Climate_Change_on_groundwater_recharge_in_Ho_Chi_Minh_City_Area_Vietnam, 2015.*

*Tu, T. A., Tweed, S., Dan, N. P., Descloitres, M., Quang, K. H., Nemery, J., Nguyen, A., Leblanc, M., and Baduel, C.: Localized recharge processes in the NE Mekong Delta and implications for groundwater quality, Sci. Total Environ., 845, 157 118, https://doi.org/10.1016/j.scitotenv.2022.157118, 2022.*

**• L95: as the statement is written, it says that extreme events, like a typhoon, would have an effect on the astronomical tides. However, they do not alter this response.**

Thank you for your remark. Indeed, we will remove "astronomical tide" and restructure this sentence accordingly.

**• Figure 3: all the components in the diagram are talked about in the main text of the manuscript, with the exception of the "mapping and characterization". The authors should explain this more. Also, on the figure label, the focus is on the "hydrological system", but it also talks about coastal processes. I recommend changing the wording toward "estuarine system" which implies both coastal and hydrologic processes.**

Thank you for your remark. The authors will provide a more detailed explanation of the "mapping and characterization" component mentioned in the diagram. This component refers to the analysis of the different sources of information and results in order to characterize the impact of this typhoon on compound flooding in HCMC and upstream of HCMC and explain the interaction of the different drivers. We will explain this more clearly in the next version. Additionally, the wording in the figure label will be changed to "estuarine system" to reflect the inclusion of both coastal and hydrologic processes.

**• L110: the authors lack a justification for the selection of a 3-day rainfall total for this analysis. All the datasets have a maximum daily time scale. Why not select a daily accumulation rather than a 3-days total? Also, the word "adequate" needs a quantification. What is adequate for the authors might not be for other readers.**

Thank you for your remark. Indeed, we did not justify our choice. The reason is that this is the time frame of the heavy precipitation event which starts on the 24[th] and ends on the 26[th.] . Additionally, using the 3-days total instead of a daily total allows us to mitigate some daily missing data in the in-situ data that we obtained from the HCMUDC - HOS in this data-scarce region. The selection of a 3-day rainfall total for the analysis will be justified in the manuscript for clarity of the reader.

**• L112: What criteria the authors used to "deem sufficient" the quality of the observed data?**

Thank you for your remark. Indeed, we do not provide an extended explanation on how we selected the rain gauges used for the study. We choose rain gauges data mainly based on the availability of the data. Some gauges presented meaningful gaps of several weeks with no data. These gauges were not used for the study. Only rain gauges with more than 450 daily measurements during the selected period (about 3 years) were used. This amount of measurements was chosen given that during the dry season HCMC experiences very few rainy days. We will clarify this in the revision.

**• Table 1: the nomenclature for the correlation coefficient equation is missing. What represents "cov(P,O)"?**

Thank you for your remark. The nomenclature for the correlation coefficient equation will be included, specifying the meaning of "cov(P,O)" to be the covariance between the predicted and observed values.

**• Table 3 is in the text before being cited. The table should be cited first and then shown. Also, how can the authors visualize a semi-diurnal tidal behavior if the time resolution of the tidal gauge has a daily time step, meaning only one value per day?**

Thank you for your remark. The table will be cited first before being presented in the text in the next revision. Regarding the visualization of semi-diurnal tidal behavior, there is a mistake in Table

3. The time resolution of the tidal gauge is hourly and not daily. This will be corrected in the revision.

**• L173: what was the time window for the moving average performed for the monthly tide values?**

Thank you for your remark. In order to remove the monthly variability in water level time series, we subtract the monthly moving average (window size is equal to 30 days) from the hourly tide gauge or 10-minute river gauge water level time series. We will clarify this in the next version.

**• L177: mention the amount of tidal constituent used in the resynthesize analysis.**

Thank you for this remark. We use all the constituents (146) except the 6 constituents that include quasi-periodic meteorological effects thus, the total amount of constituents is 140. This will be mentioned in the revision.

**• L185: generally, you should not refer to a figure before presenting other ones. For example, the authors cite Figure 7, but only have presented three figures.**

Thank you for this comment. The reference to Figure 7 will be removed to ensure other figures are presented before it.

**• L216-217: the authors should justify why they used the selected thresholds of dH and dt.**

Thank you for this comment. Indeed we did not justify this in this paragraph. These values come from a calibration step which is only mentioned in L221-L228. We will restructure this paragraph such that the source of these values is immediately clear to the reader.

**• L260-261: have other studies found similar results with ERA5?**

Thank you for this question. At the time of writing the authors did not find any comparable studies using ERA5 in similar regions. However, a thorough literature research will be done and a comparison and discussion of these results versus other previous results will be included in the discussion section of the revised version.

**• L303-304; L320-322: are these findings also been found by other researchers? Find additional literature that supports or refutes your findings. That should be part of your new discussion section.**

Thank you for this remark. Similarly to the answer to the previous remark, other findings will be compared and contrasted with other this work, supported by additional literature, which will be incorporated into the new discussion section. At the moment, no other findings directly related to the area of study have been found.

**• Figure 6: why the observed discharge is higher in the wet season than in the dry if the observed water level is higher in the dry season than in the wet? Discharge is computed from the water level, so they should have the same behavior, which is not the case.**

Thank you for this remark. The main driver of discharge as estimated via our method is the slope between the two river stations. During the dry season the water levels at both stations tend to be higher on average than in the wet season but this difference in magnitude on the seasonal average is not necessarily transferred to the instantaneous slope of the water surface. As discussed in the text (L476-480), a proportional change in water level at both stations leaves the slope variable constant and thus, the discharge is little affected. Therefore, it is physically possible that we have stronger slopes with lower seasonal average water levels and estimate higher average seasonal

discharge. This strong dependence on slope is one of the drawbacks of our method to estimate discharge as discussed in the Discussion section. We will clarify this in the revision.

**• L343: where the coastal water level is the main driver and not the rainfall?**

Thank you for this question. In this sentence we are referring to the Saigon river water levels at least all the way upstream to our Phu Cuong river gauge where we clearly see these effects. The logic behind it comes from the statistical interpretation of our data: in the rainy season, we have very clearly much stronger precipitation, yet the river water levels are lower in this season which, at first, might seem paradoxical. On the other hand, the coastal water levels decrease in this season due to the change in direction of the monsoon wind. Hence, at the seasonal scale the river water levels are controlled by the coastal water levels. This is explained in text in L311-315. We will make it clearer that we see evidence that this downstream control is valid for the whole extension of the Saigon river.

**• L381-387: move out from results into methods and data collection. This will explain to the reader why the authors also consider wind data. It was quite strange when I saw wind vectors in Figure 2.**

Thank you for this remark. We will displace this paragraph to the methods section, providing a justification for its inclusion. The placement of wind vectors in Figure 2 will then be more clear.

**• Figure 7: add a legend to the figure explaining each color of the lines. Also, add the datum to which the levels are referenced.**

Thank you for this remark. The color of the lines is already explained in the first sentence of the caption of Figure 7: black is the results for the Phu Cuong location, grey for Thao Dien and blue for Vung Tau. Nonetheless, we will make it more visible in the revision.

There are no datums that can be used as reference for the water level measurements in the river. The tide gauge is the only one to have a station datum as provided in the repository of the Sea Level Center of the University of Hawaii (link to Vung Tau station datum information: https://uhslc.soest.hawaii.edu/stations/?stn=383#datums . However, given the unknown datum of the river stations we cannot compare them. In order to mitigate this problem with perform mean normalization across the gauges such that the tidal signal is fluctuating about zero, as mentioned in the response to Reviewer 1. This allows the comparison between gauges. We will clarify this in the revision.

**• L410-424: these are not results and more a description of the study area. I would move them out and into the study area section, including the figures. Maybe the wind vector panels in figure 2 can be swapped with the top three panels in Figure 8.**

Thank you for this remark. We will take it in consideration when revising the manuscript.

**• Figure 8: panel a) the track line in the legend is green but in the map is purple. Panel b) add the datum of the elevation from the DEM.**

Thank you for this remark. Indeed, the DEM datum information is not present in the legend. The SRTM vertical datum is global mean sea level and is based on the WGS84 Earth Gravitational Model (EGM 96) geoid as specified in:

1. U.S. Geological Survey, Earth Resources Observation and Science (EROS) Center. (2018). USGS EROS Archive - Digital Elevation - Shuttle Radar Topography Mission (SRTM) 1 Arc-Second Global. Retrieved from https://www.usgs.gov/centers/eros/science/usgs-eros-archive-digital-elevation-shuttle-radar-topography-mission-srtm-1

We will take implement these changes in the revised version.

**• Conclusion: Add a paragraph about the limitation/assumption the method used by the authors may have.**

Thank you for this remark. We will implement a new paragraph discussing the limitations of our approach and methods.

**Thank you for your feedback and valuable suggestions regarding the manuscript. We appreciate your insights and will address each of your concerns in the revision.**

---

## Author Response (AR1)

In this document we present the authors response to both anonymous reviewers. However, we would like first to clarify a few points:

- We have re-structured the paper, as asked by the reviewers, by separating our previous "Results and Discussion" section into a "Results" section and a "Discussion" section. Hence, in the marked-up version of the manuscript the full "Discussion" section appears as brand new material (text is in blue). However, about 80% of this section was already written in the original manuscript and it was simply moved to the discussion section.

- The largest amount of new material can be found in the introduction section. As requested by the reviewers, we have done a thorough literature review and situated our study in the field of compound flooding assessment.

- In the marked-up version of the manuscript there are some layout problems (references coming out into the margin, extra spaces and figures that are slightly displaced). However, these problems are due to latexdiff automatic formatting and are not present in the revised manuscript document.

We highly appreciate and are very thankful for the time and effort that was invested in reviewing our manuscript. We would like to thank both reviewers for initiating this discussion. After carefully studying the constructive queries and comments, we have thoroughly revised our manuscript in an attempt to clarify as much as possible its content. Below you will find our comments (in blue) to your feedback (in **black, bold**).

**RESPONSE TO ANONYMOUS REVIEW #1**

**General comments:**

**In this paper the authors describe the impact of Typhoon Usagi rainfall on the hydrology around Ho Chi Minh City, Vietnam on 25th November 2018. In particular they evaluate the impact of intense rainfall and storm surge, on the hydrological system using tools to characterize these coastal vs surface runoff ('continental') contributions. The authors contrast the river, surface runoff and coastal (tidal) responses at two river gauge stations upstream of the Typhoon-struck coastline. The preprint is well referenced and structured, with good quality figures and tables to support the key messages. The hypothesis is given, but objectives have not been stated clearly. References are occasionally missing to back up statements in the text. Some technical terms and conclusions should probably be explained further for a general audience to avoid confusion. Some of the technical aspects in the method (and conclusions drawn from this) were difficult to follow so perhaps would benefit from being clarified/rewritten (e.g. surface runoff assumptions downstream of a dammed area). The standard of English is good, the title reflects the contents of the manuscript also. While the abstract could more concisely describe method and results, in the main body of the paper the methodology, the presentation of results, and the conclusions reached are all satisfactory. I have not checked the statistics contained in the appendix tables - these are accepted 'as is'. The manuscript requires some editorial assistance from NHESS (some grammar errors noted).**

**However, the approach taken in this paper is an interesting one, and therefore I believe the manuscript will ultimately contribute something new to the scientific discourse. I recommend accept subject to (quite a few) minor revisions as described below:**

Thank you very much for your contribution. The aspects to be improved found in your comment were considered in the revised manuscript. The objectives of the paper are stated more clearly in addition to the hypothesis. Overall, we will take your comments below and implement them in order to make the draft as clear as possible in the revised manuscript.

**Minor revisions:**

**While the Aim of this manuscript is clear (to investigate the precipitation and storm surge impacts from Typhoon Usagi on the local hydrology of the Saigon river, HCMC), the manuscript would benefit from having the objectives clearly stated in the introduction section too (L46-51).**

Thank you for this comment. We have reformulated the introduction to state more clearly the objectives of the paper. The concerning part reads as follows:

**L104-L108:**

*"The objectives of this paper are i. to provide an observation-based, multi-approach methodology to characterise the drivers of compound flooding after an historical TY, ii. to better understand which of the potential contributors to urban flooding (rainfall-runoff, storm surge or river flood) were most relevant during this particular event and iii. to characterize how the different parts of the hydrological system (terrain elevation, land cover, precipitation, tidal river and coastal surge) contribute to the response of the hydrosystem."*

**It would benefit the paper to be clearer with terminology, from the beginning of the manuscript, and to use it consistently throughout. Some examples:**

**1. A cleaner differentiation between river levels and sea levels (and river gauge vs tide/sea gauge stations). The phrase 'water levels' is a little generic even when discussing data from around a tidally influenced river /estuary.**

Indeed. We have standardized the terminology throughout the manuscript. Thank you.

**2. What is H / water level? It is not stage (with a datum from the river gauging station), seemingly. Is it depth of water above the (unknown) channel bed level, or head?**

Thank you for your comment. We utilize the term 'water level ($H$)' to refer to the height of the column of water above the pressure gauges in the river. The tidal oscillations are propagated from the coastal tide gauge to the river gauges. This is the signal that is captured by the gauges and that we call 'water level ($H$)'. Since there is no fixed datum between river and tide gauges, we normalize all signals by mean removal. This makes the tidal harmonics to oscillate about zero for all gauge locations thus, making them comparable. In addition, the $dH$ parameter in the equation to estimate discharge (Eq. 2 in the manuscript) provides an additional calibration parameter that helps mitigate this problem. We made this clearer in the manuscript as follows:

**L284-L293:**

*" (…) with the water level Hup and Hdn measured at Phu Cuong (PC) and Thao Dien (TD), respectively, and L the distance between the two locations. **We use the term 'water level (H)' to refer to the height of the column of water above the pressure gauges in the river. The tidal oscillations are propagated from the coastal tide gauge to the river gauges. Since there is no fixed datum between river and tide gauges, we normalize all signals by mean removal. This makes the tidal harmonics to oscillate about zero for all gauge locations thus, making them comparable. In addition, the dH parameter in the equation 2 provides an additional calibration parameter that helps mitigate this problem. The term dt is a time lag required to account for the propagation of the tidal wave between one location to the other**. The full observed water levels at Phu Cuong and Thao Dien were used to compute the total discharge of the Saigon river. The predicted tidal signals obtained via harmonic tidal analysis at these stations were used to compute the discharge due solely to tidal fluctuations. Then, the tidal discharge is subtracted from the total discharge in order to obtain a residual discharge - the discharge due to non-tidal effects."*

**3. In section 3.6: Water discharge is a phrase that doesn't translate well - do the authors mean river (fluvial) discharge?**

Yes, we mean river (fluvial) discharge. We now use this terminology in the revised manuscript. Thank you.

**It would benefit the paper to support particular statements with more references. E.g.,:**

**- L 30 "Vietnam lies within the most active cyclogenesis regions in the world".**

Thank you for this remark. The western North Pacific which includes the South China Sea is the most active basin of cyclone activity in the world. We will further support this statement using a selection of the following references:

1. Gao, S., Zhu, L., Zhang, W. et al. Western North Pacific Tropical Cyclone Activity in 2018: A Season of Extremes (2020). Science Reports 10, 5610 https://doi.org/10.1038/s41598-020-62632-5
2.
   Klotzbach, P. J., Wood, K. M., Schreck, C. J., Bowen, S. G., Patricola, C. M., & Bell, M. M. (2022). Trends in global tropical cyclone activity: 1990–2021. Geophysical Research Letters, 49, e2021GL095774. https://doi.org/10.1029/2021GL095774
3.
   Feng, X., Klingaman, N.P. & Hodges, K.I. Poleward migration of western North Pacific tropical cyclones related to changes in cyclone seasonalit (2021). Nat Commun 12, 6210. https://doi.org/10.1038/s41467-021-26369-7
4.
   Ruifen Zhan, Ming Ying, Peiyan Chen, On Tropical Cyclone Activity Over the Western North Pacific in 2012 (2013), Tropical Cyclone Research and Review, https://doi.org/10.6057/2013TCRR01.04.

L25-L26:

*"Vietnam lies within the most active 25 cyclogenesis region in the world **(Gao et al., 2020; Feng et al., 2021; Klotzbach et al., 2022)** and 4 to 6 TYs (TYs) hit the coast every year from October to December (Thuan et al., 2016)."*

**- L38 & L74-76. HCMC is one of the most vulnerable coastal regions in the world to flooding: Why does it rank most vulnerable (More of a certain type of flood hazard than other LECZs? A greater population at risk? More likely to /higher frequency of flooding than other locations?)? It has already been stated that the probability of typhoon occurring in southern Vietnam is not large (L33).**

Thank you for your comment. Ho Chi Minh city is often presented as one of the most vulnerable cities in the world with respect to climate change. Some of these vulnerabilities are water-related issues such as lack of urban services like drinking-water management, sanitation and rainwater drainage, In particular, flooding vulnerability is linked to sea level rise, rainfall intensification and ground subsidence, which can reach 0.02 m/year (in some geological areas), while 65% of the city is located at less than 1.5 m above sea level. In addition, HCMC is home to almost 10 million inhabitants and its population grows at about 3% per year with these risks posing a threat to many livelihoods. The urban growth rate (about 16 $km^2$ per year since 2000) is also an important factor as imperviousness of soils reduces infiltration potential and increases the flood risk (UNESCO Water, megacities and global change: portraits of 15 emblematic cities of the world, 2016).

Several studies that consider HCMC as a hotspot of vulnerability to climate change can be found in literature such as

1. Nicholls, R. J. (1995). Coastal megacities and climate change. GeoJournal, 37(3), 369-379.
2. Dasgupta, S., Laplante, B., Meisner, C., Wheeler, D., & Yan, J. (2007). The impact of sea level rise on developing countries: a comparative analysis. Climatic Change, 93(3-4), 379-388.
3. Nicholls, R. J., Wong, P. P., Burkett, V. R., Codignotto, J. O., Hay, J. E., McLean, R. F., … & Woodroffe, C. D. (2007). Coastal systems and low-lying areas. Climate Change 2007: impacts, adaptation and vulnerability. Contribution of Working Group II to the Fourth Assessment Report of the Intergovernmental Panel on Climate Change.
4. Carew-Reid, J. (2008). Rapid assessment of the extent and impact of sea level rise in Viet Nam.
5. Webster, D., & McElwee, P. (2009). Urbanization dynamics and policy frameworks in developing East Asia.
6. ADB (2010). The economics of climate change in Southeast Asia: a regional review.
7. Birkmann, J., Garschagen, M., Kraas, F., & Quang, N. (2010). Adaptive urban governance: new challenges for the second generation of urban adaptation strategies to climate change. Sustainability Science, 5(2), 185-206.
8. Fuchs, R.J.(2010) Cities at Risk: Asia's Coastal Cities in an Age of Climate Change
9. Fuchs et al.(2011) Floods in Megacities: A case study of vulnerabilities and response capacities in Metro Manila 10.
10. Hanson et al.(2011) A global ranking of port cities with high exposure to climate extremes

**- L47-48 Perhaps introduce the concept of/your meaning of the terminology "coastal and continental effects".**

Indeed, a formal explanation of our understanding of "coastal" versus "continental" effects is missing. All typhoon-related phenomena that influences coastal dynamics (such as storm surge, wind, tide) is referred to as "coastal" effects. On the other hand, typhoon-related phenomena influencing the continental hydrology (precipitation, surface run-off, river discharge, flooding) is referred as "continental' effects. Thank you for your comment, we provide this explanation in the caption of Fig. 3 for clarity. It reads as follows:

Caption Figure 3:

*"Framework to characterize and study the compound flood drivers brought by TY Usagi on the estuarine system. All typhoon-related phenomena that influences coastal dynamics (such as storm surge and wind) is referred to as "coastal" effects. Typhoon-related phenomena influencing the continental hydrology (precipitation, surface run-off, river discharge, flooding) is referred as "inland' effects."*

**- L48-50 the sentence beginning "For the first time in a data scarce region, satellite and in-depth measurements were gathered and jointly analyzed during an unprecedented extreme event ..." might require some qualification for two reasons. Firstly there are gauges and data as shown in Fig 1 (is data scarce because it is incomplete?). Secondly, more generally, there are a number of papers that have combined satellite data with (limited) data collected on the ground in areas which are considered 'data-sparse' and this is often explored through the lens of extreme flood events as case studies. E.g. Dung et al., 2011 (https://hess.copernicus.org/articles/15/1339/2011/), Kuenzer et al., 2013 (https://www.mdpi.com/2072-4292/5/2/687), Mohammed et al., 2018 (https://www.mdpi.com/2072-4292/10/6/885), Tegos et al., 2022 (https://www.mdpi.com/2306-5338/9/5/93 ). Perhaps it is just sentence construction -i.e., it's the first time this new method has been applied, in a "data scarce" region?**

Thank you for this comment. You are correct in that it is a sentence construction issue and the literature provided is very relevant. The message we would like to put across is that the methodology has never been used in such a region (to the best of our knowledge) but also, that the use of multi-source data has never been used in this specific basin namely, the Saigon river

basin. We use the term "data-scarce" to refer to the fact that it is very difficult to obtain reliable, free data in this region given that it exists. We have re-structured the introduction section and this comment has been taken into account..

**- L72-74. The Trinh et al., 2020 reference I believe refers to the wider Northwest Pacific Ocean being one of the most active regions of the world for Tropical Cyclones [TCs] (~30% of all annual tropical storms), not the South China Sea region. Many of the NWP TCs don't travel into this smaller area. It would be beneficial to clarify/correct this statement.**

Thank you for this comment. Indeed it is the Northwest Pacific Ocean which includes the South China Sea (also known as East Sea of Vietnam) that is referred as the most active region in the world. This was corrected in the text of the revised manuscript:

**L137-L139:**

*"The Saigon River estuary where HCMC is situated borders the East Sea of Vietnam (also known as South China Sea)* **which is part of the Northwest Pacific Ocean,** *one of the most active TY basins in the world with about 30 \% of the world's annual tropical storms occurring in this region (Trinhet al. 2020)."*

**– L 74. Please define a typhoon (e.g., wind speeds or category scale) vs a tropical storm.**

Thank you. We use the terms tropical storm, severe tropical storm and typhoon according to the intensity classification of the Japan Meteorological Agency who officially monitors tropical cyclones that occur within the Northern Hemisphere between the anti-meridian and 100°E. The definitions are based on 10-min average maximum wind speed as follows:

| | |
|---|---|
| Tropical Depression | Maximum wind speed < 17m/s (34kt) |
| Tropical Storm | 17m/s (34kt) ≤ Maximum wind speed < 25m/s (48kt) |
| Severe Tropical Storm | 25m/s (48kt) ≤ Maximum wind speed < 33m/s (64kt) |
| Typhoon | 33m/s (64kt) ≤ Maximum wind speed < 44m/s (85kt) |
| Very Strong Typhoon | 44m/s (85kt) ≤ Maximum wind speed < 54m/s (105kt) |
| Violent Typhoon | 54m/s (105kt) ≤ Maximum wind speed |

We will not add this information to the integral text since we believe that it is not essential for the understanding of our study by the reader.

**– L127. Technically there are four categories in Table 2, not three.**

Thank you. This has been corrected in the new version:

**L197-L198:**

*"In this analysis five datasets that belong to the* **four** *categories were chosen in order to examine their different performances over HCMC."*

**– L168. Please define 'low net discharge' – i.e. low is relative to what/under what categorization?**

Thank you for this comment. 'Low net discharge' is in comparison with the instantaneous discharge due to tidal fluctuations which are one order of magnitude higher than the net discharge. We will precise this in the revised manuscript:

**L238-L239:**

*"The Saigon river discharge is highly influenced by the mixed, semi-diurnal tidal cycle and presents a relatively low net discharge (Nguyen et al., 2019b, 2020; Camenen et al., 2021)* **which is one order of magnitude lower than the instantaneous discharge.** *"*

**– L216. dH is introduced to correct for an unknown datum. How was it derived/calculated?**

This parameter was derived by using a non-linear least-squares curve fitting of equation 1 to two 24 hour ADCP discharge measurements (as in Camenen et al. 2021). In short, we use this technique to find the best fitting possible between equation 1 and discharge measurements while taking into account measurement uncertainty. This effectively minimized the Root Mean Square Error (RMSE) between ADCP measurements and estimated discharge. This is mentioned in the revised manuscript:

**L286-L289:**

*"**The tidal oscillations are propagated from the coastal tide gauge to the river gauges. Since there is no fixed datum between river and tide gauges, we normalize all signals by mean removal. This makes the tidal harmonics to oscillate about zero for all gauge locations thus, making them comparable. In addition, the dH parameter in the equation 2 provides an additional calibration parameter that helps mitigate this problem.**"*

1. Camenen, B., Gratiot, N., Cohard, J. A., Gard, F., Tran, V. Q., Nguyen, A. T., Dramais, G., van Emmerik, T., & Némery, J. (2021). Monitoring discharge in a tidal river using water level observations: Application to the Saigon River, Vietnam. Science of the Total Environment, 761, [143195]. https://doi.org/10.1016/j.scitotenv.2020.143195

**- datums generally are unstated throughout this paper?**

Thank you for this comment. As briefly mentioned before, there are no datums that can be used as reference for the water level measurements in the river. The tide gauge is the only one to have a station datum as provided in the repository of the Sea Level Center of the University of Hawaii (link to Vung Tau station datum information: https://uhslc.soest.hawaii.edu/stations/?stn=383#datums . However, given the unknown datum of the river stations we cannot compare them. In order to mitigate this problem with perform mean normalization across the gauges such that the tidal signal is fluctuating about zero, as previously mentioned and already corrected in the revised manuscript.

**What is the mean sea level reference datum - Is that local mean or global mean? Also, in L237-240 – the datum could be provided for the SRTM DEM; this is relevant if (river and coastal) flood levels are measured against these elevations.**

Indeed, this information is not present in the text. The SRTM vertical datum is global mean sea level and is based on the WGS84 Earth Gravitational Model (EGM 96) geoid as specified in:

1. U.S. Geological Survey, Earth Resources Observation and Science (EROS) Center. (2018). USGS EROS Archive - Digital Elevation - Shuttle Radar Topography Mission (SRTM) 1 Arc-Second Global. Retrieved from https://www.usgs.gov/centers/eros/science/usgs-eros-archive-digital-elevation-shuttle-radar-topography-mission-srtm-1

Throughout the manuscript when using the term "mean sea level" we refer to the global mean sea level used as datum for the SRTM data. This information has been added in the revised manuscript:

**L310-L315:**

*"This data is currently distributed free of charge by USGS and is available for download from the National Map Seamless Data Distribution System, or the USGS ftp site. The vertical error of the DEM's is reported to be less than 16m (Farr et al. 2007).* **The SRTM vertical datum is global**

*mean sea level and is based on the WGS84 Earth Gravitational Model (EGM 96) geoid. Throughout the manuscript when using the term "mean sea level" we refer to the global mean sea level used as datum for the SRTM data."*

**- L221-224 – the introduction of K, dt and dz parameters is difficult to understand without context, perhaps rephrase this paragraph to clarify why they are important, what they mean and how they are used to optimise RMSE if this is important to your manuscript.**

Thank you for this comment. We introduced and precise the importance of these parameters for the discharge estimation in the revised manuscript. The manuscript now reads:

**L295-L301:**

*"During the asymmetric tide the equation has much more difficulty following the discharge measurements than during the symmetric tide.* **The parameters to be calibrated are as follows:**

**- K, the Manning-Strickler coefficient of the river reach is a measure of channel roughness or friction and is assumed constant.**
**- dt, is a time lag required to account for the propagation of the tidal wave between the downstream location to the upstream location.**
**- dz (which is a mistake and will be modified in the revised manuscript to read 'dH'), is used to compensate for the fact that the reference points of each location are different and unknown."**

**- Unclear about the statement that river slope explains seasonal variation in discharge rates (L344-346) and lack of discharge response after intense rainfall (L475-479). Perhaps explain a bit more the thinking in these sentences. [Hup-Hdn] should be relatively constant if levels at both locations change by approximately the same amount?**

Thank you for this comment. In both L344-346 and L475-479 (original manuscript), the argument is similar. We have re-written the explanations to make them more clear:

**L563-L567:**

*"Several arguments could hold:* **firstly, we see that river levels are generally lower during the wet season due to the coastal forcing. However, the estimated river discharge, which is a function of the slope, is higher in the wet season despite lower river levels. We propose that this is explained by a decrease in river water level everywhere in the river such that the slope is less influenced than the local river levels by this coastal forcing. Thus, we still capture seasonal differences in discharge given that our discharge estimate is a direct function of river slope. "**

**L624-L628:**

**"The relatively weak hydraulic response could be explained by the fact that both stations have a quasi-simultaneous increase in water level which leads to a less steep slope and thus, weaker discharge. This is due to the regional scale of the Usagi-brought precipitation. Hence, the slope should remain relatively constant. Thus, a wide-spread precipitation event such as TY Usagi can cause increased water level at both our measuring stations simultaneously making it challenging to accurately use this model to estimate discharge"**

**L559-560- "…high tide removing possibility of surface runoff to the river". I don't understand this reasoning/sentence. Please explain? Do you mean obscuring the response?**

Thank you for this comment. From figure 7c) and figure 9b), we can see that the peak precipitation coincides with peak river discharge. Additionally, both of these coincide with an asymmetric tide period where high tide is followed by a high-water low tide (see figure 7b). What we mean in this sentence by "removing possibility of surface runoff" is that the river water levels were high and rainwater would not effectively drain towards the river but rather linger in the impervious streets of HCMC. In fact, during spring-tides it is common to have river-induced, short-lived flooding in the lower elevation areas of the city, namely in Thao Dien. Adding to this an extreme, persistent precipitation caused wide-spread flooding. So, we propose that river high water is delaying the surface run-off and prolonging the flood residence time. We changed this in the manuscript:

**L683-L689:**

*"Time lag in river surge and peak precipitation are due to the high tide removing the possibility of surface runoff to flow towards the river.* ***The river water levels were high and rainwater would not effectively drain towards the river but rather linger in the impervious streets of HCMC. In fact, during spring-tides it is common to have river-induced, short-lived flooding in the lower elevation areas of the city, namely in Thao Dien. Adding to this an extreme, persistent precipitation caused wide-spread flooding. Hence, river flood is delaying the surface run-off and prolonging the flood residence time.*** *This shows that despite the lack of storm surge the coastal tidal forcing is still a main player in the dynamics of urban flooding in HCMC even during TY Usagi."*

**Fig A1 – colorbar units have been cut off.**

Noted. This was corrected in the revised manuscript. The units now read 'mm'.

Thank you for your comments and corrections, which were very helpful to ensure the correctness of this paper. The invested efforts are much appreciated.

**RESPONSE TO ANONYMOUS REVIEW #2**

**Summary**

The manuscript evaluates the compound inundation in Ho Chi Minh City in Vietnam during Typhoon Usagi. Their main purpose was to determine which flood mechanism drives the flood along the Saigon-Dong Nai river system. The analysis was performed by analysing observed data and remote-sensing products. Their finding suggests that the estuary system is mainly dominated by coastal processes, despite the fact that the typhoon event only brought rainfall inundation.

**General Comments**

The manuscript presents a challenging problem to assess in a data-scarce region subject to extreme hazard events, especially the interaction between coastal and hydrologic processes. I have mixed feelings if this manuscript has "enough novelty" to be accepted in a peer-review journal. Since the authors did not develop a new technique or method to investigate the proposed issue, and the results are very specific to this region. Thus, they do not either present broader and general results for the region. However, the compound flood assessment is in high demand, and this manuscript could be a good resource in the literature once it goes under a major revision.

First, there is a lack of novelty in the manuscript, not in the approach selected. In the current version of the manuscript, the novelty of applying the skewness of surge to determine which flood driver dominates it is not highlighted enough, for example. Like this, several other components of the methods are "novel enough" to be published but need to get more attention in the introduction. Thus, I highly recommend including a literature review in the manuscript that summarizes other studies that have used similar techniques to the authors and identify the missing gaps of previous works and how this manuscript tries to fill them.

Second, the authors should focus the theme of the manuscript on a "compound flood assessment" rather than a study of the hydraulic/hydrology response of the watershed. The authors are underselling their work and should put more emphasis on the "hot topic" of compound floods, which is, in reality, what the authors are doing since they are also considering coastal processes and their impacts. I strongly suggest rewriting and refocusing on this theme, including the title.

Third, the manuscript format can be improved substantially to follow a "storytelling" rather than a report. For example, the authors have a "Results and Discussion" section, but a "Discussion" section follows this one. The "Discussion" section is, in reality, a sub-section of the results since they focus on the flood impacts at the urban center, whereas the discussion section should be for comparing their results with previous findings and the physics. The authors did a great job discussing their results in the "Results and Discussion" section. Thus, I strongly recommend separating the discussion from the "Results and Discussion" section and making it a stand-alone section called "Discussion". In addition, the current "Discussion" section should be a sub-section on the new "Results" section.

Lastly, there needs to be a more coherent nomenclature and wording with the current published studies within this field. This could be from a translation from their native language to English. For example, the authors used the word "continental" to refer to hydrologic effects on the flood. However, current studies use the word "inland" more to differentiate from the coastal process in a compound flood event. Thus, the authors assess the "inland and coastal effects" on the hydrosystem, not the "continental and coastal effects". Similarly, the word "evacuate" is being used oddly for the field when referring to the riverine water leaving its banks and flooding the community. Also, the authors used the term "extreme water levels events", whereas the community uses more "extreme flood events".

Our initial motivation for writing this manuscript was in light of the previous research conducted by Camenen et al. in 2021 (introduced in L41-L45 of the original manuscript). In this research, which

provided a monthly evaluation of the Saigon River's response to this extreme rainfall, there emerges a paradoxical result of a lack of direct response in both water level and discharge. However, the authors of the current manuscript recognize the importance of investigating the finer-scale behaviour of the hydrosystem during and immediately after this event, as the monthly average may not capture the dynamics adequately. This is the reason why we chose our title to be more generally focused on the hydrosystem. However, as correctly pointed out by Reviewer 2, the manuscript addresses a challenging problem of assessing compound inundation in a data-scarce region. We do believe that the most important outcome of this manuscript becomes exactly this one. We appreciate the suggestion to focus the theme of the manuscript on a "compound flood assessment" rather than solely on the hydraulic/hydrologic response of the watershed. You are correct that our study addresses the issue of compound floods by considering both coastal and hydrologic processes. We will reframe the manuscript to put more emphasis on the compound flood assessment, as this is a hot topic in the field. Additionally, we will change the title for the final revision to be: *"Assessing Typhoon-induced compound flood drivers: a case study in Ho Chi Minh City, Vietnam"*.

We understand the reviewer's concerns regarding the novelty of the study. While we did not develop a new technique or method, we believe our research contributes to the field by applying and adapting existing approaches to the context of this case study. We would like to clarify why we strongly believe the results provided in this manuscript are valuable for the scientific community. As requested by the reviewer wa have done a thorough literature review and situated our study. This has been done in the introduction of the revised manuscript:

**L53-L103:**

[revised manuscript text omitted]

Thank you for pointing out the need for improved manuscript formatting and a clearer storytelling approach. We agree that the current structure could be enhanced to provide a more coherent flow of information. We made the necessary changes by separating the current "Discussion" section from the "Results and Discussion" section and creating a standalone "Discussion" section. The original "Discussion" section is now incorporated as a subsection in the new "Results" section. This revision will ensure a more logical organization of the manuscript and allow for better comparison of our results with previous findings and underlying physical processes.

We appreciate the comment regarding the nomenclature and wording used in the manuscript. We apologize for any confusion caused by the terminology inconsistency. We revised the manuscript to use the more commonly accepted term "inland" instead of "continental" to refer to hydrologic effects on the flood, aligning with current studies in the field. Similarly, we replaced the term "evacuate" with a more appropriate term to describe the riverine water leaving its banks and flooding the community. Additionally, we modified the phrase "extreme water levels events" to "extreme flood events" to align with the standard terminology used in the community.

**Specific Comments**

**• L24: remove the word "coastal" from "coastal engineers" since it can also help water resources engineers. I will also remove the word "reliable forecasting" there is a lot of effort needed to get to this point, such as computational resources, meteorological forecast inputs, accurate models, and not just the basic understanding of the hydrodynamics of the system.**

Thank you for your remark. This sentence has been removed from the revised manuscript as it was found that it is not essential for the introduction.

**• L25: researchers almost never do decision-making activities, as this statement suggests.**

Thank you for your remark. This sentence has been removed from the revised manuscript as it was found that it is not essential for the introduction.

**• L35: give an example of population density from another major city (e.g., New York, Hong Kong, Mumbai, etc.) so the reader can have a fair comparison for this statement.**

Thank you for your remark. To provide a fair comparison, we provide the example of population density from Paris, France to enhance the reader's understanding. We have also specified that this density refers to the urban city center. It reads as follows:

**L34-L35:**

"(…) a mega city with a population density that can reach up to 30,000 inhabitants/$km^2$ **in the urban city center** (Nguyen et al., 2019b) **(for comparison, Paris has a density of about 20,000 inhabitants/km²)** ."

**• L55: be consistent with your acronyms. The authors first used LECZ to refer to a low-elevation coastal zone, but in this statement did not use the acronym. Similarly happens with HCMC throughout the entire manuscript.**

Thank you for your remark. Consistency in acronyms has been ensured throughout the manuscript. The authors have consistently used "LECZ" to refer to the low-elevation coastal zone and "HCMC" for Ho Chi Minh City.

**• L60: quantify the "short spatial scale". Give an example.**

Thank you for your remark. Here we refer to short spatial scale to mean the scale of the district size of HCMC namely in the order of magnitude of the kilometer. For example, it can be that it is raining heavily in one district whereas in another one it is not raining at all. We will clarify this in the revision. It reads as follows:

**L117-L118:**

"For a given heavy rain event, the precipitation shows a high variability at short spatial scales *(in the order of the km)*"

**• L64: describe what it means to have a negative discharge value on this gauge.**

Thank you for your remark. Having a negative discharge at this location means we have river flow towards upstream. It reads as follows:

**L121-L122:**

"(…) and high water discharge from the Saigon river (between -1500 m³s⁻¹ and 2000 m³s⁻¹, Camenen et al. (2021), *where negative discharge represents river flow towards upstream)*. "

**• L68: Where the tides dominate in the river? Until what river length from the outlet or it is complete?**

Thank you for your questions. Indeed, the tide dominates the totality of the river Saigon from the confluence with the Dongnai river to the outlet of the Dau Tieng reservoir. We rephrase this sentence to read:

**L123-L125:**

*"Tidal fluctuations dominate the totality of the river Saigon from the confluence with the Dongnai river to the outlet of the Dau Tieng reservoir. Thus, affecting both its water levels and discharge, with regular flooding in low-lying urban districts during high spring tides."*

**• Figure 1: Need to add a map that shows where HCMC is within Vietnam and then zoom into the basin and the city. Panel (a) add the label for the Vietnam-Cambodia border and the name of the main rivers. What are the grey lines in panel (b)? need to add it to the legend.**

Thank you for your remarks. A map depicting the location of HCMC within Vietnam has been added. The grey lines are depicting the complex natural and artificial canal network around the main rivers. The legend has been updated to explain this:

Caption of Figure 1:

*"(a) The estuary of the Saigon-Dong Nai river system (blue). **Grey lines represent the complex, natural and artificial canal network of the area.** The Saigon river watershed is divided into four sub-catchments: upper, middle, lower and urban. (b) The Ho Chi Minh City center (grey) and the location of surrounding gauges. The water level gauges (orange triangles) were used for the harmonic tidal analysis and the rainfall gauges (red circles) were used for validation of gridded precipitation datasets.The Pham Van Coi rainfall gauge and the Vung Tau gauge are located outside the spatial extent of (b). The area shown in (b) is represented by the red box in (a). **(c) Map of Southeast Asia where the area in red is represented by the red box."***

**• L85-91: the authors give too many details about the classification of the typhoon in this paragraph. I would condense this since it is not pertinent to the manuscript.**

Thank you for your remark. We have condensed that paragraph and it now reads:

**L150-L154:**

*"In Fig. 2 the track of TY Usagi near HCMC is shown using the lowest pressure as indicator. The Joint Typhoon Warning Center (JTWC) assessed its intensity to be equivalent to Category 2 status on the Saffir–Simpson scale. On November 25th, the JTWC downgraded Usagi from TY to a tropical storm as central convection weakened. Usagi made landfall on Vung Tau, Vietnam at 07:00 UTC as a tropical storm, with the JTWC downgrading Usagi to a tropical depression later that day."*

**• L94: did the authors consider soil type? They only have datasets of topography and land use, but they talk about infiltration and groundwater recharge as one of the main processes during the flood but do not talk anything about the soil types which govern these processes.**

Thank you for your remark. Indeed, while datasets of topography and land use are available, the discussion is expanded towards the consequences of soil type rather than the soil type itself. Namely, we discuss infiltration and groundwater recharge during the flood (Khai et al. 2015, Tu et al. 2022).

*Khai, H. Q. and Koontanakulvong, S.: Impact of Climate Change on groundwater recharge in Ho Chi Minh City Area, Vietnam, In proceedings: THA 2015 International Conference on Climate Change and Water & Environment Management in Monsoon Asia, https://www.researchgate.net/publication/275643904_Impact_of_Climate_Change_on_groundwater_recharge_in_Ho_Chi_Minh_City_Area_Vietnam, 2015.*

*Tu, T. A., Tweed, S., Dan, N. P., Descloitres, M., Quang, K. H., Nemery, J., Nguyen, A., Leblanc, M., and Baduel, C.: Localized recharge processes in the NE Mekong Delta and implications for groundwater quality, Sci. Total Environ., 845, 157 118, https://doi.org/10.1016/j.scitotenv.2022.157118, 2022.*

**L593-L601:**

*"However, we found evidence of widespread flooding immediately east of the watershed around the river Vam Co Dong which has similar land cover as the middle part of the watershed (Fig. 8c). Another possible driver of the lag time between rainfall and discharge are aquifer recharging processes. In the middle part of the watershed the groundwater is more influenced by rainfall than by river recharge with shallow aquifers being predominantly recharged by heavy wet season rainfall events (Tu et al., 2022). The existence of a time lag between rainfall and discharge at this location after such heavy rainfall might be due to the recharge of the groundwater table functioning as a buffer. Additionally, monitored groundwater levels are generally above the Saigon river's water level creating an hydraulic gradient from the shallow aquifers towards the river (Khai and Koontanakulvong, 2015; Tu et al., 2022). This possibly indicates a groundwater recharge phenomenon followed by a slow spill towards the river over the few days following the event."*

**• L95: as the statement is written, it says that extreme events, like a typhoon, would have an effect on the astronomical tides. However, they do not alter this response.**

Thank you for your remark. Indeed, we removed "astronomical tide" and restructured this sentence accordingly. It now reads:

**L156-L158:**

*"The impact of a TY on compound floods in an estuarine system can be separated into inland and coastal drivers (Fig. 3). The first in the form of rainfall-runoff, infiltration and exfiltration influenced by topography and land use and the second in the form of storm surge."*

**• Figure 3: all the components in the diagram are talked about in the main text of the manuscript, with the exception of the "mapping and characterization". The authors should explain this more. Also, on the figure label, the focus is on the "hydrological system", but it also talks about coastal processes. I recommend changing the wording toward "estuarine system" which implies both coastal and hydrologic processes.**

Thank you for your remark. The authors will provide a more detailed explanation of the "mapping and characterization" component mentioned in the diagram. We added a sentence that reads as follows:

**L166-L169:**

*"Then, these signals are used to estimate the discharge of the river Saigon throughout the event. Lastly, the mapping and characterization step encompasses the analysis of the different sources of information and results in order to characterize the impact of this TY on compound flooding in HCMC and upstream of HCMC and explain the interaction of the different drivers."*

Additionally, the wording in the figure label was changed to "estuarine system" to reflect the inclusion of both coastal and hydrologic processes and rephrased as follows:

Caption Figure 3:

*"Framework to characterize and study the compound flood drivers brought by TY Usagi on the estuarine system."*

**• L110: the authors lack a justification for the selection of a 3-day rainfall total for this analysis. All the datasets have a maximum daily time scale. Why not select a daily accumulation rather than a 3-days total? Also, the word "adequate" needs a quantification. What is adequate for the authors might not be for other readers.**

Thank you for your remark. Indeed, we did not justify our choice. The reason is that this is the time frame of the heavy precipitation event which starts on the 24[th] and ends on the 26[th.] . Additionally, using the 3-days total instead of a daily total allows us to mitigate some daily missing data in the in-situ data that we obtained from the HCMUDC - HOS in this data-scarce region. The justification in the paper now reads:

**L175-L177:**

*"Given the time frame of the heavy precipitation event which started on November, 24th and ended on November, 26th we chose the cumulative precipitation over 3 days as the adequate time resolution for the period of available observed precipitation data (2016-2018). Additionally, using the 3-days total instead of a daily total mitigates some daily gaps in the in-situ data."*

**• L112: What criteria the authors used to "deem sufficient" the quality of the observed data?**

Thank you for your remark. Indeed, we do not provide an extended explanation on how we selected the rain gauges used for the study.  The sentence has been modified to read:

**L177-L181:**

*"The evaluation was performed over 9 rain gauge locations chosen based on the availability of the data. **Some of the rain gauges presented data gaps of several weeks and were not used for the study. Only rain gauges with more than 450 daily measurements during the selected period (about 3 years) were used. This amount of measurements was chosen given that during the six months of dry season HCMC experiences very few rainy days.**"*

• **Table 1: the nomenclature for the correlation coefficient equation is missing. What represents "cov(P,O)"?**

Thank you for your remark. The nomenclature for the correlation coefficient equation will be included, specifying the meaning of "cov(P,O)" to be the covariance between the predicted and observed values. The caption now reads:

Caption Table 1:

*"Equations and optimal values of statistical indices. P_i and O_i denote predicted and observed values, respectively, of precipitation on the i^{th} day. **cov(P, O) denotes the covariance between the predicted and observed values.**"*

• **Table 3 is in the text before being cited. The table should be cited first and then shown. Also, how can the authors visualize a semi-diurnal tidal behavior if the time resolution of the tidal gauge has a daily time step, meaning only one value per day?**

Thank you for your remark. The table is now cited first before being presented in the text. Regarding the visualization of semi-diurnal tidal behaviour, there is a mistake in Table 3. The time resolution of the tidal gauge is hourly and not daily. This has been corrected.

• **L173: what was the time window for the moving average performed for the monthly tide values?**

Thank you for your remark. In order to remove the monthly variability in water level time series, we subtract the monthly moving average (window size is equal to 30 days) from the hourly tide gauge or 10-minute river gauge water level time series. This is mentioned in:

**L242-L243:**

*"According to Cid et al. (2017), we first remove the average water level variability from the water level time series by subtracting the monthly moving average."*

• **L177: mention the amount of tidal constituent used in the resynthesize analysis.**

Thank you for this remark. We use all the constituents (146) except the 6 constituents that include quasi-periodic meteorological effects thus, the total amount of constituents is 140. The following sentence was added at the end of the paragraph:

**L254:**

*"Hence, we use a total amount of 140 constituents."*

• **L185: generally, you should not refer to a figure before presenting other ones. For example, the authors cite Figure 7, but only have presented three figures.**

Thank you for this comment. The reference to Figure 7 has been removed to ensure other figures are presented before it.

**• L216-217: the authors should justify why they used the selected thresholds of dH and dt.**

Thank you for this comment. Indeed we did not justify this in this paragraph. These values come from a calibration step which is only mentioned in L221-L228. This paragraph has been such that the source of these values is immediately clear to the reader:

**L294-L306:**

*"In Camenen et al. (2021), the model calibration is done using two ADCP campaigns: i. March 2017 during an asymmetric tide and ii. September 2016 during a symmetric tide. During the asymmetric tide the equation has much more difficulty following the discharge measurements than during the symmetric tide. The parameters to be calibrated are the following:*

*– K, the Manning-Strickler coefficient of the river reach is a measure of channel roughness or friction and is assumed constant.*

*– dt, is a time lag required to account for the propagation of the tidal wave between the downstream location to the upstream location.*

*– dH, is used to compensate for the fact that the reference points of each location are different and unknown.*

*The parameters K, dt and dH are calibrated one at a time to optimize the Root Mean Square Error (RMSE) which provided good results. However, in this study we improve this calibration by using a non-linear least squares fitting technique (not presented here). The optimal parameter values found for this study were dH = -0.149 m and dt = -2 h. This calibration method yielded better results for the estimation 305 of discharge than in Camenen et al. (2021). We improve the RMSE of total discharge during an asymmetric tide from 350 m3s−1 to 185 m3s−1 using K = 27 m1/3s−1, dt = 2 h and dH = -0.15 m."*

**• L260-261: have other studies found similar results with ERA5?**

Thank you for this question. We added a paragraph in the discussion session that reads as follows:

**L545-L558:**

*"It was found that the worst performing dataset over this domain is ERA5. ERA5 presented large values of RMSE and MBE and low linear correlation which indicate that ERA5 is overestimating rainfall over the whole domain. This finding is in line with current literature: Lavers et al. 2022 found that the largest ERA5 errors are in the Tropics and that ERA5 presents a general wet bias. Additionally, it was found that ERA5 can capture locations and patterns of extreme events but it cannot model the observed precipitation totals. Jiang et al 2021 found that ERA5 has difficulties in accurately detecting moderate and high daily precipitation events (above 10 mm/day) over mainland China. It also found that in relatively wet climate such as in the tropical climate zone ERA5 has higher RMSE than satellite - based precipitation products. Indeed, satellite-based products perform generally better than model-based products in low latitudes (Xu et al. 2021). Rivoire et al. 2021 also found that while ERA5 and CMORPH products would agree over the midlatitudes, they disagreed over the tropics. In fact, reanalysis products usually struggle to resolve precipitation over the tropics especially tropical cyclones and their surrounding environment. Slocum et al. 2022 analysis showed biases in the ERA5 environmental diagnostic quantities where the most significant discrepancies are observed in the thermodynamic fields. Notably, there is a cold temperature bias in the boundary layer, which constrains convective instability. Additionally, ERA5's biases in temperature are evident in the upper troposphere and are accompanied by a notable overestimation of relative humidity."*

• **L303-304; L320-322: are these findings also been found by other researchers? Find additional literature that supports or refutes your findings. That should be part of your new discussion section.**

Thank you for this remark. As tidal waves propagate upstream from the coast into a river, they tend to lose energy due to various factors, resulting in a decrease in tidal amplitudes. There are several mechanisms that contribute to tidal attenuation in rivers: Friction and Channel Morphology, Convergence of Tidal Energy, Interaction with River Discharge, Reflection and Refraction, Resonance and Natural Frequencies, Local Topography and Bathymetry. The concept of tidal attenuation is important for understanding the hydrodynamics of estuaries and rivers, especially in the context of flood risk. The degree of tidal attenuation can vary widely depending on the specific characteristics of the river, estuary, and tidal conditions. However, no other findings directly related to the area of study have been found. Additionally, for the surge results no other literature has been found over this area. These are considered new results provided by our study.

• **Figure 6: why the observed discharge is higher in the wet season than in the dry if the observed water level is higher in the dry season than in the wet? Discharge is computed from the water level, so they should have the same behavior, which is not the case.**

Thank you for this remark. The main driver of discharge as estimated via our method is the slope between the two river stations. During the dry season the water levels at both stations tend to be higher on average than in the wet season but this difference in magnitude on the seasonal average is not necessarily transferred to the instantaneous slope of the water surface. As discussed in the text, a proportional change in water level at both stations leaves the slope variable constant and thus, the discharge is little affected. Therefore, it is physically possible that we have stronger slopes with lower seasonal average water levels and estimate higher average seasonal discharge. This strong dependence on slope is one of the drawbacks of our method to estimate discharge as discussed in the Discussion section.

• **L343: where the coastal water level is the main driver and not the rainfall?**

Thank you for this question. In this sentence we are referring to the Saigon river water levels at least all the way upstream to our Phu Cuong river gauge where we clearly see these effects. The logic behind it comes from the statistical interpretation of our data: in the rainy season, we have very clearly much stronger precipitation, yet the river water levels are lower in this season which, at first, might seem paradoxical. On the other hand, the coastal water levels decrease in this season due to the change in direction of the monsoon wind. Hence, at the seasonal scale the river water levels are controlled by the coastal water levels. This is explained in text in L?-L?. We will make it clearer that we see evidence that this downstream control is valid for the whole extension of the Saigon river. This made clear in:

L425-L426:

*"This shows that coastal water level is the main driver of river water levels and not rainfall **over the whole extension of the Saigon river.**"*

• **L381-387: move out from results into methods and data collection. This will explain to the reader why the authors also consider wind data. It was quite strange when I saw wind vectors in Figure 2.**

Thank you for this remark. The concerning paragraph has been displaced to the methods section, providing a justification for its inclusion. It now reads:

L324-L333:

*"Wind. Storm surge is produced by water being pushed onshore by the force of the winds moving cyclonically around the storm. The impact on surge of the low pressure associated with intense storms is minimal in comparison to the water being forced toward the shore by the wind (NOAA, 2023). Additionally, many other factors, such as angle of approach of the typhoon, radius of maximum winds and the slope of the continental shelf may also have an influence (Sebastian et al., 2019). Storm size also significantly contributes to the generation of storm surge (Trinh et al., 2020) and provides an indication of the spatial region influenced by the typhoon. Larger typhoons create higher storm surges and coastal inundation (Orton et al., 2015). Therefore, we use the wind field to determine wind direction and the size of the TY Usagi (Fig. 2).* ERA5 outputs for 10 m u and v wind components were used to map the approach of typhoon Usagi towards the southern 250 coast of Vietnam. This data is provided free of charge at the Copernicus Climate Data Store (https://cds.climate.copernicus.eu) with an hourly resolution and global grid of 0.25° for the period 1959 to present (Hersbach et al., 2020)."*

**• Figure 7: add a legend to the figure explaining each color of the lines. Also, add the datum to which the levels are referenced.**

Thank you for this remark. The color of the lines is already explained in the first sentence of the caption of Figure 7: black is the results for the Phu Cuong location, grey for Thao Dien and blue for Vung Tau.

There are no datums that can be used as reference for the water level measurements in the river. The tide gauge is the only one to have a station datum as provided in the repository of the Sea Level Center of the University of Hawaii (link to Vung Tau station datum information: https://uhslc.soest.hawaii.edu/stations/?stn=383#datums. However, given the unknown datum of the river stations we cannot compare them. In order to mitigate this problem with perform mean normalization across the gauges such that the tidal signal is fluctuating about zero, as mentioned in the response to Reviewer 1. This allows the comparison between gauges as previously discussed in this document.

**• L410-424: these are not results and more a description of the study area. I would move them out and into the study area section, including the figures. Maybe the wind vector panels in figure 2 can be swapped with the top three panels in Figure 8.**

Thank you for this remark. The paragraph has been relocated as indicated. However, we believe the figures are contributing to a better understanding of the text at their current location.

**• Figure 8: panel a) the track line in the legend is green but in the map is purple. Panel b) add the datum of the elevation from the DEM.**

Thank you for this remark. The track line color has been updated in figure 8.The SRTM vertical datum is global mean sea level and is based on the WGS84 Earth Gravitational Model (EGM 96) geoid as specified in:

1. U.S. Geological Survey, Earth Resources Observation and Science (EROS) Center. (2018). USGS EROS Archive - Digital Elevation - Shuttle Radar Topography Mission (SRTM) 1 Arc-Second Global. Retrieved from **https://www.usgs.gov/centers/eros/science/usgs-eros-archive-digital-elevation-shuttle-radar-topography-mission-srtm-1**. The DEM datum information is now present in text and in the caption:

**L309-L315:**

*"The topography maps were obtained from the NASA Shuttle Radar Topographic Mission (SRTM) 90 m DEM Digital Elevation Database. The SRTM mission has provided digital elevation data (DEMs) for over 80 \% of the globe. This data is currently distributed free of charge by USGS and*

*is available for download from the National Map Seamless Data Distribution System, or the USGS ftp site. The vertical error of the DEM's is reported to be less than 16m (Farr et al. 2007). **The SRTM vertical datum is global mean sea level and is based on the WGS84 Earth Gravitational Model (EGM 96) geoid (EROS 2018). Throughout the manuscript when using the term "mean sea level" we refer to the global mean sea level used as datum for the SRTM data**."*

Caption figure 8:

*"(a) The Saigon - Dongnai system, TY Usagi trajectory and the Saigon watershed. The watershed is split into 4 parts: upper, middle, lower and urban; (b) Digital Elevation Model (DEM) of the region. **The SRTM vertical datum is global mean sea level and is based on the WGS84 Earth Gravitational Model (EGM 96) geoid (EROS 2018)**; (c) Land use map of the region; (d-f) MSWEP daily precipitation over the three days of heavy precipitation connected to TY Usagi."*

• **Conclusion: Add a paragraph about the limitation/assumption the method used by the authors may have.**

Thank you for this remark. We re-structured our conclusion section and implemented a new paragraph discussing the limitations of our approach and methods. It reads:

L690-L705:

**"The methodology presented in this paper encompasses a data processing and analysis work applied to a complex, urban estuarine system. Its foundation lies on the correct choice of TY compound flood drivers and gathering of relevant data in order to characterize the response of the hydrological system to an extreme event. This methodology could easily be applied to any other urbanized estuary both in South East Asia and elsewhere in the world by tailoring the choice of impact factors to the region of interest. Additionally, the extreme event need not be a TY but could be any other event that provokes compound flooding and impacts the respective communities. However, in this study fortunate circumstances allowed us to observe the impact of TY Usagi on the Saigon river system, as our sensors were actively recording during its landfall. River water level measurements at such high time resolution are not generally available in data-scarce regions. Hence, efforts to work together with local researchers and authorities in order to develop data monitoring strategies is crucial for studies of this type in other data-scarce regions. A way to mitigate this problem is by using open-access data such as global precipitation datasets. However, this type of data comes with inherent data quality issues and uncertainties (Scheiber et al. 2023). Another limitation of this study is the adapted Manning-Strickler equation used to estimate discharge which, even though previously validated for the Saigon river (Camenen et al. 2021), assumes uniform flow for a tidal river. Additionally, if both river water level stations feel an increase in water level the equation will not translate this t-o increased discharge as the surface slope remains constant. Nevertheless, the equation behaves rather well representing the river discharge throughout the monitoring time and during TY Usagi."**